# Relevance of Molecular Pathology for the Diagnosis of Sex Cord–Stromal Tumors of the Ovary: A Narrative Review

**DOI:** 10.3390/cancers15245864

**Published:** 2023-12-15

**Authors:** Alexis Trecourt, Marie Donzel, Nadjla Alsadoun, Fabienne Allias, Mojgan Devouassoux-Shisheboran

**Affiliations:** 1Service de Pathologie Multi-Site—Site Sud, Centre Hospitalier Lyon Sud, Hospices Civils de Lyon, 69310 Lyon, France; alexis.trecourt@chu-lyon.fr (A.T.); marie.donzel@chu-lyon.fr (M.D.); nadjla.alsadoun@chu-lyon.fr (N.A.); fabienne.allias-montmayeur@chu-lyon.fr (F.A.); 2UR 3738, Centre pour l’Innovation en Cancérologie de Lyon (CICLY), Université Claude Bernard Lyon 1, 69921 Lyon, France

**Keywords:** *FOXL2*, *DICER1*, Sertoli–Leydig cell tumor, granulosa cell tumor, molecular diagnosis, diagnostic algorithm

## Abstract

**Simple Summary:**

In the present review, we illustrate the interests of molecular pathology for establishing an integrated histomolecular diagnosis of ovarian sex cord–stromal tumors as well as its use for prognosis and treatment. We discuss the key morphological, immunohistochemical and molecular features of each entity, as well as their respective differential diagnoses. This review is organized from the predominant cell morphology to the molecular pathology, based on a practical point of view for the pathologist. Five groups are defined: (i) Group 1: predominance of fibromatous/thecomatous cells and/or stromal cells of unusual morphology; (ii) Group 2: predominance of steroid or luteinized cells; (iii) Group 3: predominance of follicular cells; (iv) Group 4: predominance of Sertoli cells; and (v) Group 5: predominance of sarcomatoid/unclassified/poorly differentiated cells. Diagnostic algorithms are proposed to differentiate each entity within the sex cord–stromal tumor category, and the contribution of molecular pathology for diagnostic purposes is discussed.

**Abstract:**

Ovarian sex cord–stromal tumors (SCSTs) account for 8% of all primary ovarian neo-plasms. Accurate diagnosis is crucial since each subtype has a specific prognostic and treatment. Apart from fibrosarcomas, stromal tumors are benign while sex cord tumors may recur, sometimes with a significant time to relapse. Although the diagnosis based on morphology is straightforward, in some cases the distinction between stromal tumors and sex cord tumors may be tricky. Indeed, the immunophenotype is usually nonspecific between stromal tumors and sex cord tumors. Therefore, molecular pathology plays an important role in the diagnosis of such entities, with pathognomonic or recurrent alterations, such as *FOXL2* variants in adult granulosa cell tumors. In addition, these neoplasms may be associated with genetic syndromes, such as Peutz–Jeghers syndrome for sex cord tumors with annular tubules, and DICER1 syndrome for Sertoli–Leydig cell tumors (SLCTs), for which the pathologist may be in the front line of syndromic suspicion. Molecular pathology of SCST is also relevant for patient prognosis and management. For instance, the *DICER1* variant is associated with moderately to poorly differentiated SLCTS and a poorer prognosis. The present review summarizes the histomolecular criteria useful for the diagnosis of SCST, using recent molecular data from the literature.

## 1. Introduction

Gonads are derived from a coelomic epithelium thickening that occurs during the fifth week of fetal development [1] and are composed of: (i) primordial germ cells that originate from the posterior epiblast, which migrate to the gonads from the yolk sac between weeks 4 and 6 [1,2,3]; and (ii) primordial sex cords and gonadal stroma, which surround the primordial germ cells and regulate their differentiation and whose precise embryological origin is still debated (coelomic epithelium, and/or mesonephros, and/or local mesenchymal cells) [1,2]. These sex cords, which are undifferentiated at the end of the sixth week, will form granulosa and Sertoli cells in the female and male embryos, respectively [2].

The initial undifferentiated step of gonadal sex cords and stroma may explain the possible occurrence of „female sex cord–stromal tumors” in males [4,5] and „male sex cord–stromal tumors” in females [6]. Sex cord–stromal tumors represent approximatively 8% of all ovarian tumors [7]. Except for the reintroduction of gynandroblastoma, the 2020 World Health Organization (WHO) classification of ovarian tumors remains largely unchanged from the previous classification [8]. Ovarian stromal tumors are usually unilateral and occur in perimenopausal women, except for sclerosing stromal tumors which are more often seen in younger patients [9]. These tumors are benign [8], apart from fibrosarcoma [10] and steroid cell tumors that exhibit malignant behavior in one-third of the cases [8]. In contrast, sex cord tumors can occur at any age [8], and may recur, sometimes with a significant time to relapse [11]. Thus, accurate diagnosis is crucial since each subtype has a specific prognostic and surgical treatment [12]. Yet, according to a recent study, these tumors represent the second category, after ovarian mucinous tumors, with the most diagnostic discrepancies between the first diagnosis made by the pathologist and the final review by the expert gynecologic pathologist [13], underlining the inherent diagnostic difficulty of these neoplasms. These diagnostic difficulties stem from the rarity of these lesions, the morphological overlap between the neoplasms of this category, and the numerous morphological differential diagnoses [8]. Although the lack of epithelial membrane antigen (EMA) immunostaining helps to distinguish these tumors from epithelial neoplasms, immunohistochemistry is not very helpful, as it does not allow the distinction between entities of this category, since all can express the sex cord–stromal immunomarkers (inhibin, calretinin, melanA, steroidogenic factor 1 (SF1), Wilms tumor 1 (WT1), and forkhead box L2 [FOXL2]), except for steroid tumors that usually do not express FOXL2 [8,14]. Thus, molecular pathology plays an important role in the diagnosis of such entities since recurrent alterations have been identified, such as the *FOXL2* variant in adult granulosa cell tumors [15] or the *CTNNB1* variant in microcystic stromal tumors [16]. In addition, these neoplasms may be associated with genetic syndromes, such as Peutz–Jeghers syndrome for sex cord tumors with annular tubules (SCTAT) [17] and DICER1 syndrome for Sertoli–Leydig cell tumors (SLCTS) [18]. In such cases, the pathologist may be in the front line of syndromic suspicion, reinforcing the importance of an integrated histomolecular diagnosis in such entities.

In the present review, we illustrate the interests of molecular pathology in establishing the diagnosis and prognosis of sex cord–stromal tumors of the ovary. The key pathological features of each entity are presented, in the light of the recent molecular advances made in this area, in order to provide tools for the pathologist to avoid diagnostic pitfalls and rule out differential diagnoses.

## 2. Design of the Present Review: From Cell Morphology to Molecular Pathology

The data collection was performed from July to October 2023, using the PubMed Central archives (NCBI) and Google Scholar research tools and the name of each entity as the keywords, without limit regarding the year of publication, but with a focus on recent molecular advances in the field. One author (A.T.) performed a first selection of the articles corresponding to the topic of the present review. In addition, an exhaustive study of the reference section of each article included herein was carried out. Then, all articles initially selected were checked independently by another author (M.D.-S.).

With the aim of helping pathologists during the diagnosis of sex cord–stromal tumors of the ovary, the present review was organized from a practical point of view, from the predominant cell morphology to the molecular pathology. This organization allows highlighting the main morphological differential diagnoses for each entity and illustrating situations in which molecular pathology is helpful and should be used. As a result, five groups of sex cord–stromal tumors have been defined, as illustrated in Figure 1:(i)**Group 1** has a predominance of fibromatous/thecomatous cells and/or stromal cells of unusual morphology. Fibromatous cells typically resemble mature ovarian stroma cells, with a monotonous spindle-shaped appearance, scant cytoplasm, and an elongated or ovoid nuclei without atypia (Figure 1A). Thecomatous cells have abundant pale/grey cytoplasm, rarely lipid-rich, with indistinct cell membranes, and ovoid to round nuclei with small nucleoli (Figure 1B), without atypia (apart from fibrosarcoma). Stromal cells may have an unusual morphology (e.g., signet ring cells; Figure 1C), often associated with fibromatous or thecomatous cells. Importantly, all cells of this group are characterized by a monocellular reticulin fiber pattern (Figure 1D).(ii)**Group 2** has a predominance of steroid or luteinized cells. Steroid cells are large, round or polygonal, with epithelioid features, abundant eosinophilic or clear/vacuolated (lipid rich) cytoplasm, with or without cytoplasmic lipochrome pigment (Figure 1E,F). Nuclei are typically round, with a central location, and a prominent nucleolus, without atypia (Figure 1G). These cytological features are those of steroid cell tumors. Importantly, luteinized cells from other neoplasms (e.g., luteinized adult granulosa cell tumor [Figure 1H]) may exhibit the same cytological features, which can be a diagnostic pitfall.(iii)**Group 3** has a predominance of follicular cells with follicle-like space formation. Adult granulosa tumor cells are polygonal, and usually have scant basophilic (sometimes eosinophilic) cytoplasm, and uniform, pale, angular to oval, longitudinally grooved, haphazardly arranged nuclei, with an irregular nuclear membrane and a “coffee bean” appearance (Figure 1I,J). Concerning juvenile granulosa cell tumors, tumor cells are polygonal and usually have an abundant eosinophilic cytoplasm, and irregular-shaped vesicular to hyperchromatic nucleus, sometimes lacking the nuclear grooves, with mild to severe nuclear atypia (Figure 1K,L);(iv)**Group 4** has a predominance of Sertoli cells. Sertoli cells are typically cuboidal or columnar cells that have a moderate amount of clear vacuolated lipid-rich to brightly eosinophilic cytoplasm, a small round nucleus, and a small nucleolus (Figure 1M–P), without atypia, especially in well or moderately differentiated tumors;(v)**Group 5** has a predominance of sarcomatoid/unclassified/poorly differentiated cells. Tumor cells are often spindle-shaped (Figure 1Q), sometimes with polygonal cells (Figure 1R), with scant eosinophilic cytoplasm (Figure 1S), moderate to severe atypia, and numerous mitoses. Importantly, the reticulin fiber pattern is not monocellular and surrounds nests of cells, differentiating them from Group 1 cells (Figure 1T).

In each group, the pathological features are presented and the relevance and contribution of molecular pathology for diagnostic purposes are discussed. When relevant, diagnostic algorithms are proposed in order to help pathologists solve difficult diagnostic situations. While the morphological classification and algorithms proposed herein could help pathologists establish a diagnosis in most cases, especially for the “classic” tumor variants/patterns, they are not exhaustive, as these are complex entities corresponding to a continuum, associating several sex cord–stromal cell types.

## 3. Literature Review Section

### 3.1. Group 1: Predominance of Fibromatous/Thecomatous Cells, and/or Stromal Cells of Unusual Morphology

#### 3.1.1. Fibroma

Fibromas are benign tumors that represent the most prevalent entity (70%) of sex cord–stromal tumors of the ovary and are classified as pure stromal tumors of the ovary [8,19]. The mean age at diagnosis varies from 42.5 to 48 years old, the tumor occurring especially in perimenopausal women [8,20]. Clinically, abdominal pain is the main symptom, sometimes associated with ascites, but the classic Meigs syndrome is rarely observed in practice [21]. This tumor can be part of a Gorlin syndrome, in which patients before 30 years old present with bilateral, multinodular, calcified ovarian fibromas, associated with multiple cutaneous basal cell carcinomas, congenital skeletal development abnormalities, and odontogenic keratocysts of the jaw [22,23,24].

Generally, except for tumors occurring within a Gorlin syndrome, fibromas are unilateral (90%), with a size ranging from 1 to >20 cm [19,20,21]. Larger tumor size is significantly associated with ascites [20]. Tumors have a lobulated surface, with white to yellow fasciculated cut sections, and calcifications are frequent, unlike necrosis and hemorrhage [8,19,25].

Classic fibromas are composed of intersecting fascicles, with a storiform arrangement [8,19]. Cell morphology is typically that of ovarian stroma: fibromatous and monotonous spindle-shaped cells, with scant cytoplasm, and elongated or ovoid nuclei without atypia [26]. In classic forms, the mitotic count is low (often ≤2 mitoses per 10 high power fields [HPFs]) [8,20]. The presence of few areas containing sheets or nodules of plump and round to spindle-shaped cells with clear/vacuolated cytoplasm resembling perifollicular thecal cells [27] could correspond to thecomatous cells, while no threshold of thecomatous cells is currently defined to diagnose fibroma vs. thecoma. Few eosinophilic hyaline globules stained with periodic acid-Schiff (PAS) [28], hyalinized plaques, edematous remodeling, or calcifications [8,21], Verocay-like features [19], bizarre nuclei [29], or even a few microscopic foci of hemorrhage may be observed [19]. While ischemic necrosis from infarction of voluminous tumors may be seen [30], the presence of tumor cell necrosis is very uncommon and should raise the suspicion of fibrosarcoma [8,19,26]. Three subtypes are described: (i) fibromas with increased cell density/cellularity [26], referred to as cellular fibromas, which account for 10% of all fibromas [19], are less often associated with edematous changes and may rarely contain melanin pigment [31]; (ii) mitotically active fibromas defined by an increased mitotic count (mean of 6.7 nonatypical mitoses per 10 HPFs; ranging from 4 to 19) [19], which may be associated with the cellular fibroma subtype [32]; and (iii) fibromas with minor sex cord elements, in which the sex cord component resembling granulosa cells or Sertoli cells represents less than 10% of the total tumor surface [32]. These subtypes must be recognized and specified in the pathological report, as they have recurrence potential [33,34], but also to avoid a misdiagnosis as fibrosarcoma in the case of cellular/mitotic fibroma, or as adult-type granulosa cell tumors or SLCTs in the case of fibroma with minor sex cord elements.

In classic, cellular, or mitotically active fibromas, the reticulin stain reveals a dense monocellular network: all cells are surrounded by reticulin fibers. However, in fibromas with minor sex cord elements, the reticulin stain shows cell cords or nests without individual reticulin fiber cell wrapping, in the areas of sex cord elements, while the surrounding fibroma shows a monocellular pattern of reticulin stain [35]. The main microscopic features of ovarian fibromas are illustrated in Figure 2.

Fibromas do not express EMA but may express sex cord–stromal immunomarkers. The expression of inhibin and calretinin, however, is often focal/weak [8], while the intensity could be higher in thecomatous or minor sex cord element areas [36].

Concerning molecular biology, the main abnormalities reported in classic fibromas are copy number alterations (CNA) in 88% of the cases, with trisomy or tetrasomy of chromosome 12 as the most prevalent CNA (63%), followed by gain of chromosomes 9 or 9q (50%), 18, and 21 (20% for each) [8,37,38], and a loss of heterozygosity (LOH) at 9q22.3 (proximal *PTCH1*) in 25% of the cases. Similarly, in cellular fibromas, LOH at 9q22.3 and at 19p13.3 (*STK11*), were found in 67% and 50% of the cases, respectively [38]. Except in Gorlin syndrome, for which *PTCH1* variants have been reported [22,23], only few gene variants have been described in fibromas: a *SMARCA4* variant associated with *PTCH1* variant in one case [24] and an *IDH1* variant associated with Ollier disease in one case [39]. This suggests that sporadic fibromas with 9q (*PTCH1* location) alterations (LOH/CNA) could arise through similar genetic pathways to fibromas associated with Gorlin syndrome. In addition, no *FOXL2* nor *DICER1* variant has been identified in fibromas [19,40]. Interestingly, the study of differential gene expression between ovarian fibromas, fibroma components of serous cystadenofibromas, and normal stroma of the ovary showed no difference [37].

In practice, molecular biology is not useful for the diagnosis of typical ovarian fibroma. For pathologists, the main purpose of molecular biology in this case would be to rule out sex cord tumors, particularly adult granulosa cell tumors and SLCTS, by searching for *FOXL2* and *DICER1* variants in the case of fibroma with minor sex cord elements and/or ambiguous reticulin stain. A diagnostic algorithm is proposed in Figure 3.

#### 3.1.2. Thecoma

Thecomas represent less than 1% of all ovarian tumors and are classified as pure stromal tumors of the ovary [8,19]. The mean age at diagnosis varies from 49.6 to 59.5 years old [41,42], the tumor occurring especially in peri- and postmenopausal women [42]. These tumors have a benign clinical behavior and patients often present with estrogenic (less commonly androgenic) manifestations, sometimes associated with endometrial neoplasms or proliferative disorders [8,41,42].

Generally, thecomas are unilateral in 95% to 97% of the cases, with a mean size of 4.9 cm [8,19,41], and measure more than 10 cm in 7% of the cases [8]. They present as yellowish solid tumors, with a lobulated and smooth surface, and rarely hemorrhagic and necrotic areas [8,41,42].

Tumor cells exhibit a diffuse arrangement or are arranged in sheets or nodules of uniform, plump and round or spindle-shaped cells [41]. Cells are composed of pale/grey cytoplasm, rarely lipid-rich, with indistinct cell membranes. The nuclei are ovoid to round with small nucleoli but without atypia, and the mitotic count is usually low [8,43]. Hyalinized plaques and sclerosis areas are classically observed [44] and are sometimes confluent giving a keloid-like appearance, with or without calcifications [8]. The use of the term “luteinized thecoma” is no longer recommended, but luteinized changes within the thecoma may be seen, with sheet/clusters of cells with abundant cytoplasm and round nuclei with a central nucleolus [19], resembling steroid-type cells [8], leading to diagnostic pitfalls. Minor sex cord elements can also be associated with thecomas [32].

The reticulin stain surrounds individual cells [43] and can be very useful, especially in distinguishing thecomas with luteinized changes from luteinized adult granulosa cell tumors, which is the most clinically important differential diagnosis, largely resolved by accurate sampling [41]. This is especially true since thecoma-like foci have been described in adult granulosa cell tumors [45,46]. When minor sex cord elements are present, the reticulin stain shows cell cords or nests without individual wrapping [8,32,35]. The main microscopic features of ovarian thecomas are illustrated in Figure 2.

Similarly to fibromas, thecomas do not express EMA. They also express sex cord–stromal immunomarkers but with a higher intensity than fibromas, especially when minor sex cord elements are present [8,36].

Only few studies have focused on the molecular biology of pure thecomas; some authors have reported a *FOXL2* variant in this neoplasm [47]. However, to date, there is some controversy about this evidence [8,19,41] as these may correspond to adult granulosa cell tumors with a predominant thecoma-like or luteinized component [45,46]. The morphological distinction between granulosa cell tumors and thecomas can be difficult and arbitrary [47]. Moreover, some authors consider that the presence of a *FOXL2* variant favors the diagnosis of adult granulosa cell tumors rather than thecomas [43]. The fact that thecomas harboring a *FOXL2* somatic variant have been reported to recur in the peritoneum with a typical morphology of an adult granulosa cell tumor indeed suggests that the initial “thecoma” might have represented in fact a luteinized adult granulosa cell tumor [14].

In practice, molecular biology is not clinically relevant if the morphology of the thecoma is typical [8]. In the case of luteinized changes in a thecoma, a thecoma with minor sex cord elements, or a thecoma with equivocal/ambiguous reticulin stain, the identification of a *FOXL2* variant would favor the diagnosis of an adult granulosa cell tumor [8,43,48].

#### 3.1.3. Fibrosarcoma

Primary ovarian fibrosarcomas are extremely rare and aggressive neoplasms, classified as malignant pure stromal tumors of the ovary [8,26]. Clinically, they typically occur in older women, aged from 41 to 76 years old, presenting with pelvic/abdominal pain or postmenopausal bleeding [10,26,49,50]. This neoplasm has also been reported in association with Mafucci, Gorlin, and *DICER1* syndromes [51,52,53], sometimes as the presenting neoplasia [53].

Generally, this neoplasm is reported as unilateral, solid, and larger than cellular and mitotically active fibromas (often >10 cm), with more surface adhesion [10,26]. Hemorrhage and necrosis are common [8].

Fibrosarcomas are hypercellular and composed of intermingled anarchic short fascicles [8,19,26]. Cells are from fibroblastic lineage [54], spindle-shaped with scant cytoplasm. Moderate to marked atypia along with high mitotic activity (>4 mitoses per 10 HPFs), are criteria to differentiate them from cellular and mitotically active fibromas [19,26]. Nuclear atypia has been found to be the only criteria associated with poor patient prognosis [55]. Thus, in the case of a high mitotic count but absence of atypia, the classification as cellular and mitotically active fibroma is recommended [8]. Epithelioid morphology has been reported [54], resembling soft tissue sclerosing epithelioid fibrosarcoma. Fibrosarcomas have also been reported in association with fibro(theco)mas with minor sex cord elements, which could suggest a continuum from the benign (fibroma/fibrothecoma) to the malignant counterpart (fibrosarcoma) [50].

The immunoprofile of ovarian fibrosarcoma is not specific [10]. Fibrosarcomas may exhibit focal staining with sex cord–stromal immunomarkers, especially inhibin, and are typically negative for CD10 [8,55,56] and positive for vimentin and smooth-muscle actin [10,52]. The expression of progesterone and estrogen receptors is variable [10], complicating the differential diagnosis with fibrosarcoma from soft tissues.

Given the rarity of ovarian fibrosarcomas, only a few studies have focused on their molecular biology. Trisomy 12 was reported in fibrosarcomas, as well as in cellular fibromas [57]. In contrast, tetrasomy 12 and trisomy 8 were reported in fibrosarcoma only, and some authors suggest that these abnormalities could be used to distinguish between cellular fibroma and fibrosarcoma [53,57]. The amplification of large regions of chromosome 8 in fibrosarcoma was later confirmed, with additional amplifications in large regions of chromosomes 1, 2, 7, and 17, with amplification of *MYC* and deletion of *TP53* [53]. *DICER1* and *NF1* variants have been identified by whole exome sequencing [53]. However, in the light of recent data, this type of ovarian neoplasm would probably be classified as undifferentiated SLCTS, or as the emerging entity called “*DICER1*-associated sarcoma” rather than fibrosarcoma [58,59,60].

In practice, *FOXL2* variant testing is not useful since the differential diagnosis is more that of a cellular/mitotically active fibroma rather than a granulosa cell tumor. In the case of fibrosarcoma suspicion, the identification of trisomy 8 could be helpful, although additional data are required to support the specificity of this molecular event. Moreover, considering the recent molecular data concerning *DICER1*-associated sarcomas [59,60], the latter diagnosis, as well as poorly differentiated SLCTS, must be considered in the case of a *DICER1* variant identification in a “fibrosarcomatous-like” neoplasm.

#### 3.1.4. Sclerosing Stromal Tumor

Sclerosing stromal tumor (SST) represents a rare and benign pure stromal neoplasm of the ovary, accounting for 2 to 6% of ovarian stromal neoplasms [61]. This entity occurs preferentially in young patients and adolescents, with a mean age at diagnosis of 29 years; postmenopausal cases, however, have been observed [62]. Although patients present with bleeding, menstrual irregularities [63], or pelvic pain [8], SST is typically not associated with hormonal manifestations [61], despite precocious puberty or association with pregnancy having been reported [64,65]. Rare cases have been reported in association with Meigs syndrome [66,67].

Generally, SST is unilateral and solid, with a mean size of 10.5 to 12.8 cm [61,63]. Tumors are well-circumscribed, with a variegated cut surface showing yellowish to white areas, admixed with edematous or myxomatous areas [65].

SST discloses a pseudo-lobular pattern in which cellular nodules are separated by hypocellular areas. The hypercellular lobules are composed of epithelioid and spindle-shaped cells and are separated by areas of hypocellular edematous/myxomatous and sclerosis collagenous stroma [8,65]. While spindle-shaped cells have fibroma cell-like appearance, epithelioid cells (luteinized cells) have clear to eosinophilic vacuolated cytoplasm, and may show prominent luteinization, especially during pregnancy, complicating the diagnosis in such cases [19,65]. Mitotic activity is usually low [62], but unusually cellular [65] and mitotically active [61] cases have been described. However, no atypical mitotic figure has been reported, and necrosis is very uncommon [8,62]. Thin-walled vessels with a hemangiopericytoma-like appearance is one of the key diagnostic-features of this entity, reported in 100% of SSTs in some case series [62]. Although sclerosis is a key feature of SST, it has also been reported in Sertoli–Leydig cell tumors, especially in the retiform component, as well as in SCTAT [8,19]. Moreover, it has recently been shown that this criterion is not specific to SST and it can also be observed in 95% of juvenile granulosa cell tumors, 50% of thecomas, 33% of microcystic stromal tumors, and in few adult granulosa cell tumors [8,44].

As SST is a pure stromal tumor, the reticulin stain surrounds individual cells but can also surround nests of luteinized cells [68]. Bundles of smooth muscle staining with SMA and desmin may be seen within the tumor since it has been suggested that SST may originate from perifollicular myoid stromal cells [69].

SST does not express EMA but expresses vimentin, SMA, desmin, and sex cord–stromal immunomarkers with a higher intensity in epithelioid/luteinized cells [65]. Estrogen and progesterone receptors are typically not expressed [63]. FOXL2 may be expressed by luteinized cells [14]. TFE3 overexpression has been reported in 78% of SSTs, especially in epithelioid/luteinized cells, while thecomas and fibromas do not express TFE3 [62]. The main microscopic and immunohistochemical features of SST are presented in Figure 4.

Studies focusing on the molecular biology of SST have shown a subpopulation of tumor cells with trisomy 12 and 7 [68,70,71], with a low level of genomic instability, a low mutational burden, and no recurrent variant [72]. Recently, *GLI2* rearrangements were reported to be the hallmark and potential molecular driver event of SST, as they were identified in 81% of these neoplasms [72]. The main common rearrangement was *FHL2::GLI2* fusion, found in 65% of the SSTs studied. Other *GLI2* rearrangements (*DYNLL1::GLI2* fusion, and other *GLI2* partner not yet identified) were found in 15% of additional SST [72]. Those gene fusions were not identified in other types of sex cord–stromal tumors nor in a database of 9950 common cancer types [72], which strongly suggests a diagnostic hallmark.

Thus, although molecular studies should not be performed in typical SST in routine practice, the identification of a *GLI2* fusion could be useful, especially in pitfall cases such as cellular/mitotically active SST (e.g., Figure 4B) or SST with prominent luteinization during pregnancy (e.g., Figure 4E).

#### 3.1.5. Unusual Morphology: Signet Ring Cell Tumors

Signet ring cell tumors are rare pure stromal tumors of the ovary and have been presumed to be a degenerative cytoplasmic change within a stromal cell tumor [8,73]. Generally, patients are peri- or postmenopausal (median age of 53 years old) and present with a unilateral mass, without specific symptoms attributable to the ovarian tumor, and no hormonal manifestation [8,19]. Tumors in younger patients have also been reported (ranging from 21 to 83 years old) [74]. After surgical removal, the absence of relapse is the rule [75].

Generally, the tumors are unilateral and confined to the ovary [75], with a median size of 5 cm (from 3 to 15 cm; mean of 5.9 cm) [74,75,76,77,78,79], although bilateral cases have been reported [79]. Tumors have a smooth surface and a solid homogenous whitish-yellow cut surface, sometimes with cystic, hemorrhage, and necrosis remodeling [8,19,75].

Histopathologically, the tumor is solid, with a diffuse growth pattern, or slightly lobulated architecture, and is composed of various amounts of signet ring cells (15 to 95%) in a fibromatous background (10 to 85%), without microcyst formation [73,75,76]. Signet ring cells have small, bland, and homogenous nuclei, eccentrically located, with occasional grooves, and no to mild atypia. Mitotic activity is usually low, but high mitotic activity up to 16 mitoses per 10 HPFs has been reported [74,75]. The abundant cytoplasm is composed of an empty vacuole, resembling Krukenberg tumor cells, but without mucin or lipid. Cytoplasms sometimes contain hyalin globules, which are degenerating erythrocytes phagocytized by the tumor cells [8,74]. Special stains such as alcian blue, mucicarmine, or PAS are negative [74]. A steroid cell tumor component has been reported in association with signet ring cell tumor [78], suggesting either a continuum between pure ovarian stromal neoplasms, or a degenerative cytoplasmic change within a stromal cell tumor.

The distinction between signet ring cell tumor and Krukenberg tumor is based on various criteria. The criteria that are evocative of Krukenberg tumor are: bilaterality, the presence of abundant necrosis and hemorrhagic areas, the presence of glands, nests, or cords of cells, lymphovascular and perineural invasion, the presence of a pseudomyxoma ovarii, and mucin (stained by PAS, alcian blue, or mucicarmine) in signet ring cell vacuoles. The main microscopic features of signet ring cell tumors are presented in Figure 5.

The reticulin stain shows a dense fiber network and surrounds individual cells [74].

The immunohistochemical profile is crucial to rule out differential diagnoses (i.e., metastases such as Krukenberg tumors). Tumor cells are usually positive for calretinin, SF1, smooth-muscle actin, and may express FOXL2, inhibin [73,78], and focally AE1/AE3, ER/PR, without nuclear expression of β-catenin or CyclinD1 positivity. EMA is the most useful marker to rule out a metastasis, since signet ring cell tumors are negative for EMA [8,73,77,79].

Although a variant of *CTNNB1* has been reported in this neoplasm, along with β-catenin, cyclinD1, and CD10 immunoreactivity [77], the presence of *CTNNB1* variant in this neoplasm is not unanimously accepted. Some authors reported that a subset of signet ring cell tumors with β-catenin nuclear positivity and/or *CTNNB1* variant likely represents a morphological variant of microcystic stromal tumor (MCST) and must be classified as such [73]. Thus, in the case of signet ring cell tumor, an extensive sampling of the tumor is now recommended to rule out MCST [73]. No variant of *FOXL2* has been reported in the cases studied, with a low mutational load [79].

In practice, immunohistochemistry and molecular biology are useful to rule out gastro-intestinal tract metastasis as well as MCST with signet ring cell changes [73], as shown in Figure 6 which details the differential diagnosis approach based on recent data from the literature [73]. The identification of β-catenin nuclear positivity and/or *CTNNB1* variant could suggest a variant of MCST or a component of MCST associated with signet ring cell tumor.

#### 3.1.6. Unusual Morphology: Microcystic Stromal Tumors

Microcystic MCSTs are rare pure gonadal stroma neoplasms, first described in 2009, presuming to derive from ovarian stromal cells [8] and occurring mostly in perimenopausal women (median age: 45 years old) [80]. Patients present with symptoms related to the tumor mass and possible menorrhagia [81], without hormonal manifestation [82]. At diagnosis, tumor blood markers are usually normal [80,83,84,85]. While the main cases reported have benign clinical behavior [86], this neoplasm has been classified as of “uncertain behavior”, since some authors reported peritoneal locations [85] or recurrences [87]. This neoplasm may occur in association with the familial adenomatous polyposis [88]. One case has been reported in association with 5q deletion syndrome encompassing the APC deletion [81].

Generally, MCSTs are unilateral, solid, and cystic neoplasms [80,87], with a mean size of 8.7 cm (from 2 to 27 cm) [80] and a smooth surface, without extraovarian spread in most cases, although exceptions have been reported [85]. The cut surface shows yellowish to white areas, and small foci of necrosis or hemorrhage have been shown in 19% of cases [8,80].

MCSTs disclose a triad of microcysts, solid cellular zones, and fibrous stroma, in varying proportions [8]. Microcystic areas are composed of small rounded cystic spaces, with a basophilic or clear content. While both microcystic and solid patterns are usually present, microcystic patterns may be predominant in 12.5% of the tumors [80]. Solid and cellular areas show corded, tubular, and nested cell arrangements [88]. This solid pattern may be misleading if predominant, which is the case for approximately 19% of the tumors. The tumor cells are epithelioid with granular eosinophilic to basophilic cytoplasms, or intracytoplasmic vacuoles, and are rarely spindle-shaped [80,88]. Nuclei are round to oval or spindle-shaped with fine chromatin, no atypia, and small or no nucleoli. Foci of bizarre nuclei with a symplastic appearance are common, reported in 62.5% of the tumors [80], and accounting sometimes for 50% of the tumor surface [83]. The mitotic activity is usually low [8], from 0 to 2 mitoses per 10 HPFs, without increased mitosis count in bizarre nuclei areas [83]. Solid and microcystic areas are delimited by paucicellular fibrocollagenous hyalinized stroma, resembling those of thecomas [80]. Cases with focal microcyst formation and prominent signet ring cell components have been recently reported, suggesting that a subset of MCSTs are misdiagnosed as signet ring cell tumor [73], the differential diagnosis of which is explained in Figure 6. The main microscopic features of MCST tumors are presented in Figure 5.

The reticulin stain highlights individual cells surrounded by reticulin fibrils [89], supporting the stromal nature of this neoplasm.

Tumor cells diffusely and intensely express WT1, FOXL2, and SF1, but are usually negative for EMA, AE1/AE3, inhibin, and calretinin. The β-catenin (nuclear and cytoplasmic), CD10, and cyclinD1 are also expressed. While tumors sometimes express androgen receptors, they are usually negative or focally positive for ER and PR [82,89]. Synaptophysin and CD56 may be expressed, but chromogranin was always negative in the cases tested [80].

The molecular hallmark of MCSTs is a somatic missense variant in the exon 3 of the *CTNNB1*, first described in 2011 [82,84,90], and present in 70.4% of the cases reported [16,82,84,85,86,91], highlighting the involvement of the Wnt/β-catenin pathway in the genesis of MCST. The other main molecular alterations reported are linked to the familial adenomatous polyposis: a somatic *APC* variant [86,90] and a 5q deletion, encompassing the APC, in the 5q deletion syndrome [81]. ACP variant has been reported as associated with *KRAS* missense variant in this neoplasm [87]. To our knowledge, *APC* and *CTNNB1* variants are mutually exclusive [73]. One case has been reported with a somatic *FANCD2* variant, a gene that encodes a protein which plays a role in DNA repair [81]. A *RET* variant has been found in a case initially classified as signet ring cell tumor, and then reclassified as MCST [73]. In contrast, no *FOXL2* or *DICER1* variant have yet been identified in MCST [82,84,89].

The morphological features of MCSTs can be misleading with SSTs, steroid cell tumors, juvenile and adult granulosa cell tumors, Sertoli–Leydig cell tumors, and yolk sac tumors [80,92]. Thus, demonstrating the *CTNNB1* or *APC* variants may be useful in practice, especially when the solid and cellular areas are predominant [19]. The absence of *FOXL2* variant does not favor the diagnosis of luteinized granulosa cell tumor. The microcystic pattern can also be misleading with cystic/sieve-like areas of female adnexial tumor of Wolffian origin (FATWO), which could have similar immunohistochemical characteristics (no EMA or PAX8 expression, focal or negative ER/PR expression, and expression of CD10 and WT1) [93]. The cystic remodeling in Sertoli–Leydig cell tumors [19] represents another pitfall. In these settings, the presence of *CTNNB1* or *APC* variants, and the absence of *DICER1* variant, may also be useful. Moreover, in the case of *APC* variant identification, it could be helpful to suggest, in the pathology report, the possible association with familial adenomatous polyposis [92].

### 3.2. Group 2: Predominance of Steroid or Luteinized Cells

#### 3.2.1. Leydig Cell Tumor

Leydig cell tumors are benign steroid cell tumors arising from Leydig cells of the ovarian hilum, accounting for 0.1% of all ovarian tumors [94] and 19% of all steroid cell tumors of the ovary [19], for which the exact pathogenesis remains unknown [8]. This tumor usually occurs in perimenopausal women with a mean age of 58 years old (ranging from 32 to 82) [95]. Patients present with androgenic manifestations and virilizing symptoms (i.e., hirsutism) due to the increase in serum testosterone level in approximatively two-thirds of cases, or, in rare cases, estrogenic symptoms [95,96].

Generally, one key feature of Leydig cell tumors is their hilar location. Tumors are usually small with a mean size of 2.1 cm and well-circumscribed [95,97]. This tumor is typically unilateral, although bilateral cases and/or an association with hilar Leydig cell hyperplasia have been reported [8,95,97]. Leydig cell tumors are solid, with a yellowish or brownish and often lobulated cut surface [8].

Leydig cell tumors exhibit well-defined borders at low magnification without a capsule. The tumor shows diffuse or lobulated cellular growth, sometimes with pseudo-glandular architecture [19]. Tumor cells are large, round, or polygonal epithelioid cells with abundant eosinophilic or clear/vacuolated (lipid-rich) cytoplasm, with or without cytoplasmic lipochrome pigment [95]. One characteristic microscopic feature of Leydig cell tumors is the clustering of nuclei, separated by eosinophilic nuclear-free zones [95]. Although true Reinke crystals have been described as characteristic of this tumor, they may be very difficult to visualize or even absent [8,19]. In contrast, paracrystalline inclusions, which have been suggested to be precursors of Reinke crystals, are frequently observed in Leydig cells [98], as illustrated in Figure 7D. Nuclei are round, with a single central prominent nucleolus, sometimes with nuclear pseudo-inclusions. Bizarre nuclei are sometimes seen. Mitotic activity is low [19]. Another key feature of Leydig cell tumors is the fibrinoid material within blood vessel walls, which is observed in one-third of the cases [95]. The stroma between tumor cells may be hyalinized, edematous, or hemangiomatous. The latter may be predominant, and an association with anastomosing hemangiomas has been recently described in a series of Leydig cell tumors [97], which is consistent with the endocrine nature of this tumor. Association with contralateral Leydig cell hyperplasia is common [8,94]. The main morphological features of Leydig cell tumors are illustrated in Figure 7.

Leydig cells intensely and diffusely express calretinin, inhibin, MelanA, and CD99 while FOXL2 is usually not expressed, which allows their distinction from luteinized cells of adult granulosa cell tumors that express FOXL2. Leydig cells do not express EMA but may be positive for keratin and androgen receptors [8,14,19,99,100,101].

The major differential diagnosis is steroid cell tumor not otherwise specified (NOS), which is usually larger and may recur in one-third of the cases [8]. In the absence of Reinke crystals, the hilar location, the small size of the tumor, the presence of nuclei clusters separated by eosinophilic anuclear areas, and the presence of fibrinoid material within blood vessel walls favor the diagnosis of Leydig cell tumor [8,19,95]. A diagnosis algorithm between entities with steroid/luteinized cells is proposed in Figure 8. Leydig cell hyperplasia is another differential diagnosis; however, Leydig cell hyperplasia is usually bilateral, and no tumor mass is generally present.

Although it has been hypothesized that Leydig cell tumors may originate from a variant of the stimulator G protein gene [102], the molecular biology of ovarian Leydig cell tumors has not yet been well-studied. Future research must focus on the pathogenesis and molecular biology of these tumors, especially to identify factors that allow the differentiation between Leydig cell tumor and steroid cell tumor NOS.

#### 3.2.2. Steroid Cell Tumor, Not Otherwise Specified (NOS) and Malignant

Steroid cell tumors NOS account for less than 0.1% of all ovarian neoplasms [103], and correspond to non-hilar steroid tumors, presumed to be of ovarian stromal cell origin, that lack the morphological features of Leydig cells previously described [8,19]. The distinction between Leydig cell tumors and steroid cell tumors NOS is crucial since approximately one-third of the latter exhibit a malignant clinical behavior, with a potential for extraovarian spread [8]. Steroid cell tumors NOS can occur at all ages, and patients tend to be younger than patients with Leydig cell tumors, with a mean age reported from 41 to 43 years old (ranging from 2.5 to 80) [8,19,104,105]. Patients usually present with symptoms related to the ovarian mass, androgenic manifestations with elevated serum steroid hormones in 41% of the cases, and rarely estrogenic manifestations or Cushing syndrome [104,105]. This neoplasm has also been described in association with von Hippel–Lindau syndrome [106,107].

Generally, most tumors are unilateral, and rarely bilateral (6% of cases). They have a non-hilar location, and are larger than Leydig cell tumors, with a mean size of 8.4 cm (ranging from 1.2 cm to 45 cm) [104]. The cut section is usually yellow to brown, solid, and lobulated. Hemorrhage and necrosis may be present, especially in malignant tumors.

Histopathologically, steroid cell tumors NOS have a diffuse cellular growth pattern or may show nests, cords, pseudo-glandular, and follicle-like structures. Tumor cells are round or polygonal, with distinct cell membranes, and have abundant granular eosinophilic cytoplasm, that may be clarified/vacuolated (lipid rich), with lipochrome pigment in one-third of the cases. The nucleus is round and central with no to mild atypia, showing a prominent central nucleolus [8,19,104]. The mitotic activity does not usually exceed 2 mitoses per 10 HPFs. Tumor diameter ≥ 7 cm (78% malignant), diffuse moderate to severe atypia (64% malignant), high mitotic activity ≥ 2/10 HPFs (92% malignant), necrosis (86% malignant), and hemorrhage (77% malignant) are features associated with a malignant behavior [103,104,105,108]. The stroma may be either scant or composed of fibromatous or hyalinized bands with large hyalinized vessels [19,109]. Associated bilateral or ipsilateral stromal hyperthecosis reinforce the hypothesis of a stromal cell origin [104]. The main microscopic features are illustrated in Figure 7.

The immunoprofile is identical to that of Leydig cell tumors with positive expression of inhibin, calretinin, MelanA, CD99, and with FOXL2 and EMA negativity [103].

Except for Leydig cell tumors, for which the differential diagnosis with steroid cell tumors NOS has already been discussed, steroid cell tumors should be differentiated from pregnancy stromal luteoma. The latter are usually <2 cm, multinodular, bilateral, and the history of pregnancy or immediate post-partum is usually very suggestive [8,110].

Similarly to Leydig cell tumors, very little is known about the molecular biology of these neoplasms. Associations with von Hippel–Lindau syndrome have been reported, suggesting that this neoplasm could enter the spectrum of this syndrome, although somatic molecular studies have not yet been performed [106,107].

#### 3.2.3. Luteinized Thecoma Associated with Sclerosing Peritonitis

Luteinized thecoma associated with sclerosing peritonitis is an extremely rare tumor [8], initially considered as a variant of thecoma and for which the histogenesis and path-ophysiology of neoplastic or non-neoplastic nature have been debated for many years [111,112]. Recent data, however, suggest a neoplastic proliferation [113,114]. Patients present with abdominal and pelvic pain, ascites, abdominal distension, and bowel obstruction, without hormonal manifestation [111,115]. Peritoneal sclerosis is the cause of death in 11% of the cases, while no recurrence or metastasis of the ovarian lesion has yet been reported [8,111]. This neoplasm occurs at any age, with a mean age at diagnosis of 36 years (from 10 months to 85 years) [111].

Generally, ovarian lesions are bilateral in 89% of the cases, with a mean ovarian neoplasm size of 9.7 cm (ranging from 2 to 31 cm), and are associated with a peritoneal fibrosing process, as observed in peritoneal dialysis or other medical conditions [111,113]. The cut section of the ovarian tumor is solid, lobulated, and pink to grey, with areas of edema and hemorrhage, with occasional small cysts [111]. Peritoneal involvements are usually described as indurated, nodular, and lobulated [111].

The lesion may either invade the entire ovary in the case of tumor size > 9.5 cm, or circumferentially involve the ovarian cortex, with an entrapment of ovarian follicles and medullary region preservation [8,19,111]. Ovarian lesions are composed of hypercellular areas, separated by zones of marked edema, sometimes with a microcystic-like appearance. The edematous areas may be predominant (90% of the tumor surface) [8,19,111]. Tumor cells are predominantly spindle-shaped with brisk mitotic activity (from <1 to 83 mitoses/10 HPFs; mean of 18), with some clusters of round cells with pale and abundant cytoplasm (luteinized cell component, lipid-rich). Peritoneal locations have a lobulated appearance, especially in the epiploon, in which the involvement of fibrous septa is often observed. The neoplasm is composed of variable cellular proliferation of “(myo)fibroblastic-like” spindle-shaped cells, with elongated nuclei with mild atypia and fibrillar pink cytoplasm. Tumor cells are separated by fibrin or edematous, myxoid, and collagenous stroma, with a sclerosing appearance, and occasional chronic inflammatory infiltrate [111,116].

A reticulin stain usually shows reticulin fibers between individual spindle-shaped cells, and Reticulin fibers surround clusters of luteinized cells [115].

Although the luteinized cell component expresses CD56, calretinin, and inhibin, these markers are not expressed or rarely and focally expressed by spindle-shaped cells, which usually express SF1 and FOXL2 in more than 90% of the cases; EMA is negative on both components [8,111,112]. Spindle cells express ER and PR in 38% and 85% of the cases, respectively, and may also express CD117 (c-kit) in 70% of the cases, highlighting the importance to rule out the diagnosis of stromal gastrointestinal tumor, which is much more frequent than the present entity. Recently, several immunohistochemical markers (MGAT5B, NCOA3, MKI67, and β-Catenin) were found to be significantly more expressed in the luteinized component of luteinized thecomas associated with sclerosing peritonitis compared to thecomas [114].

To our knowledge, only one study reported molecular biology on luteinized thecoma associated with sclerosing peritonitis [114]. The authors described an *MGAT5B::NCOA3* fusion in one case using whole exome sequencing. The fusion was further confirmed using in situ fluorescent hybridization on the luteinized component of seven cases of luteinized thecoma associated with sclerosing peritonitis. The authors showed a difference with thecoma, for which no evidence of *MGAT5B::NCOA3* fusion was found [114]. To our knowledge, this alteration has not yet been identified in another neoplasm.

In practice, molecular biology is not useful for the diagnosis of luteinized thecoma associated with sclerosing peritonitis, because only few molecular data are available. Future studies should focus on the identification of molecular alterations and on the confirmation of the *MGAT5B::NCOA3* fusion in this entity. Thus, to date, the utility of molecular biology in these exceptional cases of luteinized thecoma associated with sclerosing peritonitis is only to rule out differential diagnoses, such as other spindle-shaped proliferations localized to the peritoneum.

#### 3.2.4. Modified Morphological Features: Luteinized Changes

Adult/juvenile granulosa cell tumors, SST, and thecomas have their own chapters. In the present section, only the modified morphological features and the differential diagnoses will be discussed. Indeed, in the case of an abundant luteinized cell component, these neoplasms could belong to the Group 2 (predominance of steroid or luteinized cells) of the present study and their distinction from steroid cell tumors may be problematic.

While adult granulosa cell tumor is the most common sex cord tumor, the luteinized pattern occurs in approximately 1% of the cases and is characterized by 50 to 90% of tumor cells resembling *corpus luteum* cells (abundant eosinophilic cytoplasm, sometimes with cytoplasmic vacuolation, without typical nuclear features of adult granulosa cell tumor) [117] (Figure 7G,H). Luteinized juvenile granulosa cell tumor is also a differential diagnosis, especially when the thecoma cell component is abundant [8,19]. In contrast to SST with abundant luteinized cells and luteinized juvenile granulosa cell tumor, luteinized adult granulosa cell tumor is less often associated with pregnancy, since most patients are postmenopausal [8,19,117]. This neoplasm must be differentiated from a steroid cell tumor and a thecoma with luteinized foci. To that end, the sampling is crucial to identify non-luteinized areas of adult granulosa cell tumor. FOXL2 immunohistochemistry testing is useful to differentiate steroid cells (FOXL2 usually negative) from luteinized cells (FOXL2 positive) [8]. However, while immunohistochemistry is not helpful for the distinction between thecoma and luteinized adult granulosa cell tumor or between juvenile and adult granulosa cell tumor, the presence of *FOXL2* variant allows the diagnosis of adult granulosa cell tumor, while the *DICER1* variant favors that of luteinized juvenile granulosa cell tumors or SLCTS [8,19,46].

Concerning STT, the luteinized cell rich pattern is observed during pregnancy and may be bilateral [65]. In this case, the hypercellular regions are composed of numerous clusters, nests, or sheets of luteinized polyhedral cells with round nuclei, a prominent central nucleolus, and an abundant eosinophilic or clear cytoplasm [65] (Figure 4E). Because of its risk of recurrence compared to SST, a steroid cell tumor NOS must be ruled out using FOXL2 and TFE3 immunomarkers, which are usually expressed in SST and not expressed in steroid cell tumors NOS [14,62]. Similarly, a luteinized adult granulosa cell tumor must be ruled out, by searching for the *FOXL2* variant [15]. Finally, thecomas do not express TFE3 [62]. In addition, in the case of diagnostic difficulties, the identification of the GLI2 rearrangement allows to classify the neoplasm as SST [72].

Thecomas may also have luteinized changes [8,19], although the use of the term “luteinized thecoma” is no longer recommended [19,65] and may be confusing with luteinized thecoma associated with sclerosing peritonitis. In this case, a luteinized adult granulosa cell tumor must be ruled out, using the reticulin stain (fibrils around individual tumor cells in thecomas) [43] and the *FOXL2* variant (a variant would favor adult granulosa cell tumor with luteinized foci) [46].

### 3.3. Group 3: Predominance of Follicular Cells

#### 3.3.1. Adult Granulosa Cell Tumor

Adult granulosa cell tumors are the most common pure sex cord tumors of the ovary [8], accounting for 95% of all granulosa cell tumors, and 1–5% of ovarian tumors [19,118,119,120]. Although it can occur at any age [121], most patients are perimenopausal, usually 50–55 years old at diagnosis [119]. Patients present with abdominal pain and swelling, estrogenic manifestations with uterine bleeding, revealing endometrial hyperplasia in 29.2% of the cases or endometrioid carcinoma in 7.5% of the cases, especially after 40 years old [122], or, although less frequent, androgenic manifestations in 10% of the cases [8]. Patients may have elevated α-inhibin and β-inhibin serum levels, and the latter can be used for the follow-up to predict recurrences. Most patients are diagnosed at FIGO stage I [118]. The prognosis depends on the stage at diagnosis, the presence of a capsule rupture, an extraovarian spread, and residual tumor after surgery [8]. Approximately one-third of the patients recur, of which 50% die from the disease [119]. The time-to-relapse may be long, up to 30–40 years after the initial diagnosis [119].

Generally, tumors are unilateral (>95% of the cases) [19], with a mean size of 9.7 cm which can vary considerably (from <1 cm to 30 cm) [118,120]. Tumors are typically solid and cystic, rarely purely cystic [123], and have a yellow to grey cut section, sometimes with hemorrhagic changes [119].

Granulosa cells are arranged in an admixture of several patterns: diffuse or solid growth of cells, which is the most common pattern, microfollicular with Call–Exner bodies, which are the hallmark of adult granulosa cell tumors but are present in only 10% of tumors (small spaces containing eosinophilic/basophilic fluid, and/or apoptotic bodies, and/or hyalinized basement membrane material) [19,121], cord/trabeculae, insular pattern, macrofollicular, sarcomatoid, gyriform, moire-silk, and pseudopapillary. These follicular cells are admixed with a variable amount of fibromatous/thecomatous background [8,46]. Morphologically, the diagnosis may be challenging in many situations, especially in the case of a sarcomatoid pattern or when the fibromatous component is abundant (fibroma-like adult granulosa cell tumor), or when the tumor is entirely cystic, luteinized, and exhibits SCTAT-like features (abundant hyalinized basement membrane material within Call–Exner bodies) [8,121]. Tumor cells usually have scant, eosinophilic or basophilic cytoplasm, and uniform, pale, angular to oval, longitudinally grooved, haphazardly arranged nuclei, with an irregular nuclear membrane and a “coffee bean” appearance [8,19,121]. Granulosa cells may be luteinized with abundant vacuolated cytoplasm and nuclei lacking the typical grooves [117]. While the mitotic activity is usually low (<5 mitoses per 10 HPFs), high-grade transformations with marked atypia, bizarre multinucleate cells, and high mitotic count have been reported [19,124]. The main morphological features of adult granulosa cell tumors are presented in Figure 9.

The reticulin stain shows reticulin fibers around nests of tumor cells and may be especially useful in sarcomatoid/fibroma-like patterns to rule out cellular fibroma [35].

Tumor cells are typically positive for sex cord–stromal markers (FOXL2, inhibin, calretinin, SF1, WT1, CD56) [14,125]. Although AE1/AE3 may be positive, as well as in other stromal sex cord tumors, EMA is negative. ER and PR show a variable positivity and PR is usually more extensively positive than ER [19]. In high-grade components, p53 may have diffuse mutation-type immunoreactivity [124].

The molecular hallmark of adult granulosa cell tumors is the missense somatic point variant (C402G) of *FOXL2*, present in >95% of tumor cells, and absent in the juvenile type and in fibromas [8,19,126,127]. However, the *FOXL2* may be wild-type in a subset of adult granulosa cell tumors, especially in the case of an abundant fibromatous background [92], and in cases with purely cystic patterns [123]. This may be due to lesser tumor cell DNA compared to the fibromatous stroma. *FOXL2* is known as a marker of ovarian differentiation and has a role in the proliferation and differentiation of granulosa cells [127,128], and its alteration reduces fertility in mouse models [129]. Recently, in vivo studies demonstrated that the somatic variant c.402C > G (p.C134W) of *FOXL2* was necessary and sufficient to trigger the genesis of adult granulosa cell tumors [129]. This alteration led to dysregulation of the TGFβ pathway signaling, which is consistent with previously reported data [128]. Apart from diagnostic purposes, alteration of the *TERT* promoter may have a role in tumor progression [130,131], while transcriptomic analyses showed that relapse is not driven by transcriptomic changes [132]. Using whole exome sequencing, several non-recurrent gene variants have been reported [131]. CGH shows recurrent chromosomal imbalances: gains of chromosomes 12 and 14 in 30% of the cases, and loss of chromosome 22 in 40–50% of the cases [131]. Variants of *TP53* are reported in high-grade components associated with atypia and numerous mitoses [124,133].

In practice, the identification of a *FOXL2* variant has a diagnostic impact [48]. This research should not be performed in the case of adult granulosa cell tumors with classic morphology and immunohistochemical profile. However, the search for a *FOXL2* variant should be performed: (i) in sarcomatoid/fibroma-like patterns or in the case of an ambiguous reticulin stain to differentiate an adult granulosa cell tumor from cellular fibroma (Figure 3); (ii) in the case of luteinized granulosa cells to differentiate adult granulosa cell tumors from steroid cell tumors, or juvenile type, or thecoma (Figure 8); (iii) in the case of <10% of sex cord elements in a fibromatous background to differentiate an adult granulosa cell tumor from fibroma with minor sex cord elements (Figure 3); (iv) in the case of a macrofollicular or pseudopapillary pattern to differentiate adult granulosa cell tumors from the juvenile type; (v) in the case of a SCTAT-like pattern to differentiate an adult granulosa cell tumor from a sporadic SCTAT; and (vii) in the case of sex cord–stromal tumor NOS, for which the identification of a *FOXL2* variant would favor the diagnosis of adult granulosa cell tumor (96). Of note, *TERT* and *FOXL2* variants may be detected in circulating blood samples and are used by some teams as a non-invasive method for disease monitoring [134].

#### 3.3.2. Juvenile Granulosa Cell Tumor

Juvenile granulosa cell tumor is a very rare pure sex cord neoplasm, accounting for 5% of all granulosa cell tumors of the ovary [118,119,120]. It occurs in young women, less than 30 years old of age in 97% of the cases (mean age of 13 years old) [8], although some tumors have been reported in older women (ranging from newborns to 67 years old) [135]. Patients usually present with symptoms attributable to the pelvic mass (pelvic/abdominal pain, abdominal swelling), and estrogenic manifestations (isosexual pseudoprecocity in 82% of prepubertal patients, or menstrual irregularities, amenorrhea, or uterine bleeding in older patients), or rarely androgenic manifestations with virilization [8,19,120,135]. This neoplasm has rarely been reported in association with Ollier disease, Maffucci syndrome, DICER1 syndrome, Beckwith–Wiedmann syndrome, and tuberous sclerosis [135,136,137,138,139,140,141].

Generally, >97% of the tumors are unilateral, with a mean size of 12.5 cm (from 3 to 32 cm), and are usually solid and cystic, or predominantly solid [19,135]. The tumor cut section is white to yellowish, with areas of necrosis and/or hemorrhage in 13% of the cases [142].

Histopathologically, juvenile granulosa cell tumor displays a diffuse, lobulated, and follicle-like architecture, in a myxoid, edematous, or hyalinized background. Although round to oval macrofollicles with basophilic or eosinophilic secretions are typical features, follicle-like spaces often vary in size and shape [8,19,120,135,142]. Tumor cells usually have an abundant eosinophilic cytoplasm, and irregular-shaped vesicular to hyperchromatic nucleus that typically lacks the nuclear grooves observed in the adult granulosa cell tumor, although they are at least focally observed in approximately 50% of tumors [142], highlighting the importance of tumor sampling. Tumor cells may have mild to severe nuclear atypia, sometimes with bizarre nuclei [120]. Mitotic activity is usually brisk and higher than in adult granulosa cell tumor, from <1 to 32 mitoses per 10 HPFs (mean of 7) [135]. Areas of necrosis are reported in 40% of the cases [142]. Some clusters of thecomatous cells may be observed, leading to diagnostic difficulties with other steroids/luteinized neoplasms, especially when the tumor has luteinized changes (e.g., during pregnancy). The main morphological features of juvenile granulosa cell tumors are presented in Figure 9.

The reticulin stain shows reticulin fibers around nests/cords of granulosa cells, and a monocellular network in the thecomatous areas [19].

Similar to the adult type, tumor cells express sex cord–stromal markers (SF1, inhibin, calretinin, WT1, FOXL2). Although AE1/AE3 may be expressed, EMA is typically negative [8,14,19,120,135].

Molecular studies performed on juvenile granulosa cell tumors showed the absence of *FOXL2* variant and the presence of *DICER1* variants in 6–23% of the cases [126,143], sometimes associated with DICER1 syndrome [139]. Activating variants of *AKT1* have been reported in 29–60% of the cases and may play a central role in the pathogenesis of this neoplasm; variants of *GNAS*, a gene involved in cellular proliferation and tumor invasion, have been reported in 30% of the cases [143,144,145,146,147]. The genomic landscape of juvenile granulosa cell tumors has been recently studied, highlighting several other recurrent gene variants in different gene categories: epigenetic modifiers (*KMT2C* [52%], *ARID1A* [39%], *KMT2D* [35%]), homologous recombination (*BRCA2* [35%], *ATM* [26%], *BRCA1* [23%]), mismatch repair (*POLE* [23%], *MLH1* [19%], *MSH6* [10%]), and telomere elongation (*TERT* variant in 19% of tumors, *TERT* rearrangement in 13%) [143]. Somatic *IDH1* and *IDH2* variants have been found in cases of juvenile granulosa cell tumors associated with Ollier disease or Maffucci syndrome [8,144]. One case of juvenile granulosa cell tumor has been reported in association with germline variants of *TP53* and *PTEN*, usually observed in Li–Fraumeni and Cowden syndromes, respectively. However, only a somatic LOH of *PTEN* was found while no somatic alteration of *TP53* was observed [140], questioning the role of these syndromes in the tumorigenesis. Since then, *TP53* somatic variants have been reported in up to 16% of juvenile granulosa cell tumors [143]. Apart from their use for diagnostic purposes, the presence of *DICER1* and *GNAS* variants could worsen the prognosis, compared to wild type *DICER1* and *GNAS* tumors [126,143,144,145,146,147], although additional data are required to confirm the prognostic role of these variants.

In practice, in the case of an unusual morphology such as luteinized cells, the molecular pathology may help to rule out differential diagnoses: *FOXL2* variant would favor a luteinized adult granulosa cell tumor while *DICER1* variants could be seen in juvenile granulosa cell tumors. Also, small cell carcinoma of the ovary of hypercalcemic type shows similarities with juvenile granulosa cell tumors (young patients, brisk mitotic activity, pseudo-follicles) and should be ruled out by performing EMA and BRG1 immunostainings, the latter being a very good surrogate for *SMARCA4* alteration [148]. In sex cord tumors, juvenile granulosa cell tumors are currently considered as follicular cell-derived neoplasms. However, because of the absence of common molecular alterations found in adult granulosa cell tumors, juvenile granulosa cell tumor may represent a distinct entity, with clinical, histopathological, and molecular differences compared to common follicular cell-derived neoplasms (i.e., adult granulosa cell tumor) [126]. Future studies should focus on identifying the cellular origin of juvenile granulosa cell tumor in order to better classify this entity.

### 3.4. Group 4: Predominance of Sertoli Cells

#### 3.4.1. Pure Sertoli Cell Tumor

Pure Sertoli cell tumors are extremely rare neoplasms, defined as pure sex cord tumors, mostly composed of hollow or solid Sertoli tubules [8]. This neoplasm occurs at any age, with a mean age at diagnosis of 30 years old (ranging from 2 to 79) [149,150]. Patients present with symptoms related to the tumor mass (palpable abdominal mass, abdominal distention, abdominal/pelvic pain), and hormonal manifestations such as isosexual pseudoprecocity or vaginal bleeding in prepubertal girls, or menometrorrhagia due to endometrial hyperplasia or less commonly androgenic manifestations (virilization, hirsutism) in women of reproductive age or postmenopausal patients. Tumors may also be an incidental finding [149,150]. Some tumors have been reported with aldosterone or renin secretions [151,152], revealed by resistant arterial hypertension. In the largest series reported yet, pure Sertoli cell tumors were associated with Peutz–Jeghers syndrome in 11% of the cases [150,151].

Generally, tumors are unilateral, usually solid, less commonly solid and cystic, and rarely predominantly cystic, with a tan to yellow cut section and a size ranging from 0.8 to 30 cm (mean of 8.6 cm). Some tumors have a lobulated appearance. Areas of necrosis and hemorrhage are uncommon, present in less than 10% of the cases [149,150].

Sertoli cell tumors are typically well-delimited and may show several architectures, with hollow or solid tubules (sometimes closely packed) resembling prepubertal testicular tubules (Figure 10B) being the most frequent pattern observed, explaining that testicular feminization is one of the differential diagnoses; trabecular, alveolar, pseudopapillary, retiform, or diffuse arrangements may also be seen. One case was reported with endometrioid-like morphology [150], which must lead pathologists to rule out endometrioid carcinoma with sex cord-like formations [153]. Tubules or cords are composed of cuboidal or columnar cells that usually have moderate amounts of clear vacuolated lipid-rich to brightly eosinophilic cytoplasm (oxyphilic variant, most frequent in Peutz–Jeghers syndrome), with small round nucleoli without atypia and a mitotic count < 2 mitoses per 10 HPFs [19,150]. However, bizarre nuclei or marked cytological atypia and brisk mitotic activity (up to 24 mitoses per 10 HPFs) may be rarely observed [150]. A basement membrane-like hyalinized stroma may be associated between tubules or trabeculae [149]. The presence of few Leydig cells reported in 11% of the cases, small foci of both adult or juvenile granulosa cell tumors reported in 5.6% of the cases, and small foci of SCTAT component reported in 2% of the cases (sometimes associated with Peutz–Jeghers syndrome [19,154]) do not exclude the diagnosis of pure Sertoli cell tumor but highlight the diagnostic difficulties of sex cord tumors as a spectrum associating different components. Features suggesting malignant behavior include a tumor diameter >5 cm, a mitotic count of ≥5 mitoses/10 HPFs, nuclear atypia, and necrosis [8,19,150]. The main microscopic features of pure Sertoli cell tumors and some differential diagnoses are illustrated in Figure 10.

Tumor cells express SF1 in 100% of the cases, inhibin in 82–98%, WT1 in 96%, CD99 in 68–86%, AE1/AE3 in 65%, and calretinin in 50–60%. Less commonly, CK7, chromogranin and PR are expressed in 13% of the cases. However, EMA is always negative [150,155,156,157]. Immunohistochemistry may be useful to rule out morphological differential diagnoses other than sex cord–stromal tumors, including sex cord-like endometrioid carcinoma of the ovary (EMA+, ER+, PR+, PAX8+, inhibin−, calretinin−), carcinoid tumors since only 12% are EMA + (chromogranin+, synaptophysin+, calretinin−, inhibin−, WT1−, SF1−), and FATWO in a subset of cases (admixture of other architectural patterns, CK7+, androgen receptor+, Glutathione S-transferase+) [8,150,155,156,157,158]. Therefore, a minimal immunohistochemistry panel of EMA, inhibin, calretinin, WT1, SF1, chromogranin, synaptophysin, and CK7 seems necessary before concluding to a pure Sertoli cell tumor of the ovary.

To date, the molecular biology of pure Sertoli cell tumors of the ovary is poorly studied. The association with Peutz–Jeghers syndrome suggests a role of the *STK11/LKB1* in the tumorigenesis of a subset of tumors [150,151]. A *DICER1* variant has been identified in 63% of pure Sertoli cell tumors [159]. However, the identification of a *DICER1* variant is not specific to Sertoli cell tumors, since such a variant has been reported in other gynecological neoplasms: in 63–80% of Sertoli–Leydig cell tumors, 40% of gynandroblastomas, 6% of juvenile granulosa cell tumors, but also in *DICER1*-associated sarcomas, and rare germ cell tumors (mixed malignant germ cell tumor, dysgerminoma, yolk sac tumor, and teratoma) [22,58,59,60,126,159,160,161].

Therefore, the identification of *DICER1* variant is not mandatory to make the diagnosis of Sertoli cell tumor. However, the sequencing of *DICER1* may be useful in the case of non-typical Sertoli cell tumors, especially in cases with an association of several components (Leydig cells, granulosa cells, SCTAT component) and in cases of few tubule formations with prominent basement membrane-like hyalinized stroma. Moreover, in girls and young women, the identification of a *STK11* variant may suggest a Peutz–Jeghers syndrome and must lead to constitutional testing.

#### 3.4.2. Sex Cord Tumor with Annular Tubules (SCTAT)

SCTAT are extremely rare neoplasms accounting for <1% of all sex cord tumors [8]. They are classified as pure sex cord tumors and are supposed to derive from Sertoli cells [162]. The mean age at diagnosis depends on the association or not with Peutz–Jeghers syndrome: patients with Peutz–Jeghers syndrome tend to be younger (mean of 27 years old, ranging from 4 to 57) than patients with sporadic tumors (mean of 34 years old, ranging from 6 to 76). A Peutz–Jeghers syndrome is present in approximatively 36% of the cases [163]. While tumors typically represent an incidental finding in patients with Peutz–Jeghers syndrome, abdominal/pelvic pain, primary amenorrhea, and hormonal manifestations such as sexual precocity, menstrual irregularities, or postmenopausal bleeding may reveal the lesion [163,164]. The distinction between syndromic and non-syndromic cases is crucial, since syndromic tumors are considered benign, whereas non-syndromic tumors may exhibit extraovarian spread in 20% of the cases [8,165,166]. Patients with Peutz–Jeghers syndrome may present with other gynecological neoplasms (e.g., gastric-type endocervical adenocarcinoma, *STK11* adnexal tumor) [167,168].

Generally, patients with Peutz–Jeghers syndrome have an incidentally found microscopic tumor (not observable at gross examination) in 75% of the cases, which is bilateral in 66% of the cases. When the tumor is observed (25% of the cases; usually with a size < 3 cm), it is composed of a single or multiple nodules, with a solid and yellow cut section, and calcifications. In contrast, non-syndromic cases are unilateral, with a unique tumor mass ranging from 0.5 to 33 cm [19]; the cut section is solid, or solid and cystic (rarely predominantly cystic), and yellow, sometimes with areas of necrosis and hemorrhage [163,169].

Microscopically, SCTATs are composed of well-circumscribed round or complex nests or tubules of Sertoli cells that encircle hyaline basement membrane-like material of round or oval shape [8,19]. In these nests, nuclei are located in both the periphery of the nests and around the hyaline basement membrane-like material, spaced by anuclear cytoplasm areas (ring-like appearance or antipodal distribution of nuclei; Figure 10G–I) [163,169]. The hyaline basement membrane-like material may also surround the nests of tumor cells. Areas of Sertoli tubules, as those observed in Sertoli cell tumors, may be seen (Figure 10J), as well as small foci of solid granulosa cell tumor nests; these associations are usually reported within non-syndromic tumors [8,163,169]. In non-mass-forming tumors, occurring in association with Peutz–Jeghers syndrome, simple nests are usually distributed within the ovarian fibromatous stroma (Figure 10K,L), and associated with extensive calcifications [8,19,163,169]. Tumor cells have abundant clear vacuolated (lipid-rich; Figure 10I) to pale eosinophilic cytoplasms (Figure 10H), with small round to oval nuclei, with single small nucleoli, but without atypia, and with a low mitotic count (very rare or absent mitoses) [8,19]. However, nuclear pleomorphism and high mitotic count (>10 mitoses per 10 HPFs) may be seen in the non-syndromic setting and should suggest a malignant behavior [19,163,169].

Tumor cells do not express EMA and CD10, while calretinin, inhibin, WT1, CD56, and SOX9 are diffusely and strongly expressed [8,170,171]. FOXL2 expression is usually weak [165]. One of the main morphological differential diagnoses is gonadoblastoma, in which the sex cord component surrounding the hyaline basement membrane-like material is mixed with a germ cell component positive for SALL4. (Figure 10O) [8]. Gonadoblastoma is seen in young girls with gonadal maldevelopment, in whom the diagnosis of dysgerminoma must be ruled out [8].

Only a few studies have focused on the molecular biology of SCTAT. To date, while germline variants of the *STK11* (19p13.3) or LOH of the 19p13.3 region have been reported in Peutz–Jeghers-associated tumors, somatic variants or alterations of this gene have not yet been identified [172]. Thus, the identification of *STK11* variant in SCTAT must lead to constitutional genetic testing. Moreover, the identification of *STK11* variant in ovarian tumors is not specific to SCTAT, since it could be associated with Sertoli cell tumors of the ovary [150,151] or *STK11* adnexal tumors [158,168]. No variant of *FOXL2* [14] or *DICER1* [165] have been reported in tumors tested so far. Thus, the identification of a *FOXL2* variant must lead to reconsider the diagnosis as granulosa cell tumor with SCTAT-like features, which is one of the main diagnostic pitfalls (Figure 10M,N) [173], especially since colocalizations of SCTAT (with *STK11* variant) and granulosa cell tumors (with *FOXL2* variant) have been reported [174]. The identification of a *DICER1* variant could be helpful to differentiate SCTAT with Sertoli cell tumor-like areas (no *DICER1* variant, [165]) from Sertoli cell tumor with minor SCTAT component (*DICER1* variant in 63% of the cases [159]). Future studies should focus on the identification of the driver molecular alterations in non-syndromic SCTAT.

#### 3.4.3. Sertoli–Leydig Cell Tumor (SLCT)

SLCTs are rare tumors of the ovary, accounting for less than 0.5% of all ovarian neoplasms [175]. They are classified in the mixed sex cord–stromal tumor category [8]. This neoplasm may occur at any age (from 1 to 84 years old, mean of 25 years old), with a peak in frequency in women of reproductive age, especially before 30 years old [8,176]. The retiform pattern usually occurs in younger women (mean of 15 years old) [19]. Patients usually present with abdominal or pelvic pain related to the tumor mass, and approximately 50% of patients have a history of androgenic manifestations, such as oligomenorrhea or amenorrhea, hirsutism, voice raucity, laryngeal protuberance, and clitoromegaly [8,176,177]. Estrogenic manifestations are rarer but may concern 15% of patients [8,177]. Moderately and poorly differentiated SLCTs are well-known to be associated with DICER1 syndrome [6,178]. The occurrence of a tumor spectrum (pleuropneumoblastoma, thyroid carcinoma, multinodular goiter, rhabdomyosarcoma, medulloepithelioma, and numerous other tumors) in a young patient is highly suggestive of this syndrome [175,179,180]. Patients with well-differentiated tumors have a survival rate of 100% [8]. Indeed, the prognosis of the tumor is closely related to the tumor stage, the degree of differentiation, the mitotic count, the presence of a retiform pattern and heterologous elements, the presence of the *DICER1* variant, and the quality of the surgery [181,182]. Recurrences usually occur in the peritoneal cavity within 2 years [8,176,183,184].

Generally, SLCTs are unilateral, with a size ranging from <1 cm to 51 cm (mean of 13.5 cm) [176]. Tumor rupture at diagnosis concerns approximately 15% of the tumors [176]. SLCTs are usually solid, solid and cystic, or rarely cystic, with a white to yellow, lobulated cut section [8]. Areas of necrosis, edema, gelatinous changes, and hemorrhage may be present [19].

Histopathologically, recent data from the literature tend to separate well-differentiated SLCTs and moderately/poorly differentiated SLCTS as two distinct entities [6]. First, well-differentiated SLCTs are composed of small, round, solid or hollow, well-developed Sertoli cell tubes. Sertoli cells have the same morphological characteristics than previously described (Section 3.4.1, Pure Sertoli cell tumor), without atypia, and with a low mitotic count (Figure 11A,B); these tubes are closely admixed with a variable amount of fibromatous component containing Leydig cells organized in clusters [6,8,19,176]. In moderately and poorly differentiated SLCTS, dense lobular areas of tumor cells are usually separated by hypocellular and edematous areas, giving a multinodular appearance on low-power examination (Figure 11D) [8,19,181]. In these cases, Leydig cells may be rarer or more difficult to identify and are usually located at the periphery of the lobules. Moderately differentiated SLCTS usually have a lobular pattern, containing nests, solid tubes, or cords of Sertoli cells, admixed with a fibromatous component (Figure 11C). The tubules are less well-developed than in well-differentiated neoplasms. Microcystic features have been described [19]. Sertoli cells have mild to moderate atypia, although bizarre nuclei may be seen. A modest mitotic activity is usually present [181]. Poorly differentiated SLCTS resemble primitive gonadal stroma with sarcomatoid features, or solid tumor growth. A minor component of moderately differentiated SLCTs may be associated and must be searched to suggest the diagnosis of SLCT. Sertoli cells are poorly differentiated, with scant cytoplasm, moderate to severe atypia, and a brisk mitotic activity (Figure 11D,E); in poorly differentiated SLCTS, Leydig cells are scant, sparse, or very difficult to identify [8,19,176,183]. Moderately and poorly differentiated SLCTS may be associated with a retiform pattern (in 15% of moderately/poorly differentiated tumors), characterized by anastomosing slit-like spaces containing papillae lined by cuboidal cells resembling the rete testis (Figure 11F). Heterologous elements are present in 22–25% of moderately/poorly differentiated tumors [176,177,181,185,186] and may be epithelial (benign intestinal-type epithelium [Figure 11G] which is the most common heterologous element, but may evolve toward borderline mucinous tumor or mucinous adenocarcinoma) or mesenchymal (cartilage, skeletal muscle, rhabdomyosarcoma [Figure 11H]). Rarely, less common heterologous elements may be seen, such as a carcinoid tumor (Figure 11I) [19], hepatic differentiation [187], or neuroblastoma [186]. Although a minor component of granulosa cell tumor may be seen, this contingent must represent less than 10% of the tumor surface, or the neoplasm should better fit in the gynandroblastoma category (see Section 3.4.4, Gynandroblastoma) [19].

Using immunohistochemistry, Sertoli cells express FOXL2 in 50% of the cases [14], inhibin in 71–100% [170,188], calretinin in 57–89% [188], SF1 in 100%, and WT1 and Cytokeratin in 100%, while EMA is constantly negative [170]. Poorly differentiated tumors are sometimes negative for sex cord–stromal markers. Leydig cells have the same immunoprofile than those of Leydig cell tumors (calretinin+, inhibin+, FOXL2-; see Section 3.2.1. Leydig cell tumor). Heterologous elements usually express the same markers than their constituent tissues [8]. The use of calretinin and FOXL2 antibodies may be useful to identify rare Leydig cells (intense calretinin expression, no FOXL2 expression), especially in poorly differentiated SLCTS, the presence of which favors the diagnosis of SLCT rather than that of other poorly differentiated sex cord–stromal tumors. Moreover, the expression of ER tends to be more prevalent/intense in SLCTs compared to adult granulosa cell tumors while PR expression tends to be more prevalent/intense in adult granulosa cell tumors compared to SLCTs [189]; these markers could be helpful, especially for the differential diagnosis of poorly differentiated SLCTs and adult granulosa cell tumor, although ER/PR alone could not be used to differentiate between both neoplasms.

Concerning molecular pathology, although no recurrent alteration has been found in well-differentiated SLCTs, the *DICER1* somatic variant is highly prevalent in moderately/poorly differentiated SLCTs (approximatively 63%), associated with a germline variant in approximatively 70% of the cases [159,190]. However, *DICER1* variant is not specific to SLCT since it has also been described in other gynecological tract tumors, such as Sertoli cell tumors [159], gynandroblastomas, juvenile granulosa cell tumors, *DICER1*-associated sarcomas and rare germ cell tumors [22,58,59,60,126,159,160,161]. As a result, the use of DICER1 gene testing to differentiate SLCTS and juvenile granulosa cell tumors or gynandroblastoma is not recommended [92]. Using whole exome sequencing, a *DICER1* variant was found in only 45% of the cases of moderately to poorly differentiated SLCTS [191]. However, the authors found other recurrent somatic variants, such as variants of the *CDC27* (52.6%) and *MUC22* (21.1%); other gene variants were found in 10.5% of the cases (i.e., *MUC2*, *MUC17*, *RAD50*, *SON*, *ZNF708*, *CACNA1E*, *KIF1B*, and *PTH2*).

In practice, the diagnosis of well-differentiated SLCTs is based on morphology and does not require molecular biology, especially since no variant of *DICER1* has been found in these neoplasms [6,190]. In contrast, in the case of histopathological suspicion of moderately/poorly differentiated SLCTs, the identification of a *DICER1* variant could be used to rule out some differential diagnoses, such as sarcomatoid/poorly differentiated granulosa cell tumors, and, depending on the clinical characteristics (age, other tumor association), should lead to constitutional testing in order to not overlook a DICER1 syndrome. In addition, *DICER1* variant testing should be used to differentiate SLCTs and adult granulosa cell tumors with luteinized features: the presence of a *DICER1* variant would favor the diagnosis of SLCT [92]. Apart from its use for diagnostic purposes, the presence of a *DICER1* variant, especially in the case of the germline variant, worsens the prognosis of the patients [127,191]. Although some SLCTs have been reported with a *FOXL2* variant [14], it is not clear whether these neoplasms could correspond to poorly differentiated/sarcomatoid adult granulosa cell tumors rather than a true SLCT. Indeed, all cases of SLCT with *FOXL2* variants were seen in postmenopausal women with uterine bleeding or estrogenic manifestations, which is not the right clinical setting for SLCT but is reminiscent of clinical symptoms of adult granulosa cell tumors [8].

#### 3.4.4. Gynandroblastoma

Gynandroblastoma is an extremely rare mixed sex cord–stromal tumor with both male (Sertoli cell tumor or SLCT) and female (adult or juvenile granulosa cell tumor) components [8]. However, since the most abundant component is usually a Sertoli or Sertoli–Leydig cell component [8], this neoplasm is thus classified in Group 4 of the present study. This tumor may occur at any age, reported from 14 to 80 years old [192,193,194], with a mean age at diagnosis of 31 years old [195]. Patients present with symptoms related to the tumor mass, sometimes associated with androgenic manifestations (virilization in 42%) or estrogenic manifestations. Amenorrhea is observed in 54% of the cases [195]. Although most tumors are considered benign [8], very rare recurrences have been reported, sometimes in patients with a *DICER1* germline variant [195,196,197].

Generally, tumors are mostly unilateral with only a few bilateral cases reported [197]. The tumor size ranges from 5.5 to 20 cm (mean size of 11 cm); tumors are solid or solid and cystic, with a white to yellow cut section [8].

Histopathologically, gynandroblastomas comprise a mixture of SLCT or Sertoli cell tumor component (the latter is less frequent) with a juvenile or adult granulosa cell tumor component (the latter is less frequent) [8,192,193]. The morphological features of each component have already been fully detailed in the dedicated sections above. Each component must be significantly represented (at least 10% of the tumor surface for most authors) to consider the diagnosis of gynandroblastoma [193,195,198,199,200], especially since SLCTS or pure Sertoli cell tumors have been reported in association with minor foci of tumors resembling granulosa cell tumors, and vice versa [8,19,154,193].

The immunohistochemical profile is that of each contingent detailed above. Of note, FOXL2 is expressed in 100% of the cases, in both components [192].

Concerning molecular pathology, *DICER1* somatic variants have been identified in 19–40% of gynandroblastomas, especially in those with SLCT and juvenile granulosa cell tumor components (in both components) [159,192,201]; this somatic variant is sometimes associated with a germline variant [195,196,197]. In contrast, no variant of *FOXL2* [193] nor *AKT1* has been reported so far [192]. Although this entity was deleted from the 2014 WHO classification of female genital tumors, several arguments support a distinct entity from granulosa cell tumors or Sertoli cell tumors/SLCTS: (i) the absence of *FOXL2* variant in gynandroblastomas with an adult granulosa cell tumor-type component [193]; (ii) the absence of an *AKT1* variant in gynandroblastomas with a juvenile granulosa cell tumor-type component, which is now known to be a key driver event of juvenile granulosa cell tumors [143,144,145,146,147,192]; (iii) the presence of a *DICER1* variant in 19–40% of gynandroblastomas (lower prevalence compared to SLCT) and the presence of *FOXL2* immunohistochemical expression in 100% of the cases in both contingents (higher prevalence compared to SLCT).

In practice, in the case of a morphological suspicion of gynandroblastoma, the identification of a *DICER1* variant could reinforce this hypothesis, whereas its absence does not invalidate the diagnosis. Moreover, in view of the extreme rarity of this neoplasm, the presence of a *DICER1* variant in young patients should lead to constitutional testing for *DICER1* syndrome. A simplified diagnostic algorithm for the main entities of Groups 3, 4, and 5, based on the main molecular alterations, is proposed in Figure 12. A summary of the alterations found in sex cord–stromal tumors is given in Table 1.

### 3.5. Group 5: Predominance of Poorly Differentiated Sex Cord Cells

Few data are available regarding sex cord–stromal tumors NOS, which are defined by the absence of characteristics of a specific tumor type [8]. Indeed, sex cord–stromal tumors of the ovary are difficult to classify in approximately 10% of the cases, due to a low degree of differentiation [202,203,204]. In such cases, the differential diagnosis is usually between adult granulosa cell tumors and SLCTs. Clinical manifestations are nonspecific, related to the tumor mass (abdominal/pelvic pain, abdominal swelling), and/or to estrogenic or androgenic manifestations [8]. The mean age at diagnosis is 44 years old [205]. The 5- and 10-year survival rates are 92% and 74%, respectively, with a favorable prognosis for stage I tumors [202,206].

Generally, tumors are solid or solid and cystic, and have a grey/white to yellow cut section, sometimes with areas of hemorrhage and/or necrosis, and a median size of 6.5 cm (ranging from 0.8 to 20 cm) [8,203,204,206].

Histopathologically, although both sex cord and stromal elements are present, no distinctive feature of a sex cord entity is observed (i.e., no feature of adult granulosa cell tumor or SLCT). Historically, sex cord–stromal tumors NOS have been divided into two groups: predominance of spindle-shaped cells or predominance of cords/trabecula/tubules that do not allow definitive placement into adult granulosa cell tumor or SLCT categories [202]. Edema and the presence of Leydig or luteinized cells may be observed during pregnancy [203,206]. Tumor cells may show some atypia, with a mitotic count of up to 8 mitoses per 10 HPFs [206].

Using reticulin stain, the stromal cell component is highlighted by a dense monocellular network, while the sex cord component shows reticulin fiber wrapping nests of sex cord elements or Leydig/luteinized cells during pregnancy [8,203,206].

Immunohistochemical studies support the diagnosis of sex cord–stromal tumors, with the expression of sex cord–stromal markers such as FOXL2 immunoreactivity in 58–75% of the cases, inhibin in 75%, calretinin in 83%, SF1 in 91%; ER are expressed in 50% of the cases and PR in 67% [14,205].

By definition, sex cord–stromal tumors NOS usually display no *DICER1* nor *FOXL2* variant since they may represent a distinct entity rather than a subgroup of adult granulosa cell tumors or SLCTS [205], although one case with *DICER1* variant [160] and few cases with *FOXL2* variants have been reported [14,205]. However, the identification of these variants should lead to the review of slides and to consider the diagnoses of poorly differentiated SLCTS [159,190] and poorly differentiated/sarcomatoid/diffuse granulosa cell tumor, respectively [40], following the algorithm proposed in Figure 12, and as previously suggested [205]. In this setting, thorough sampling of the tumor is essential, to look for well-differentiated tumor areas, as well as an integrated histomolecular approach (i.e., *DICER1* and *FOXL2* variant testing) to better classify these tumors.

The molecular landscape of gynecological neoplasms is becoming more complex, with an increasing number of emerging entities defined by molecular alterations [207,208]. Among these entities, *DICER1*-associated sarcomas, sometimes presenting as a pelvic or adnexal mass, are composed of undifferentiated tumor cell proliferation, with atypia, and numerous mitoses [58,59,60]. As this emerging entity may be part of the DICER1 syndrome, located at the adnexa, composed of undifferentiated cells with rhabdomyoblastic and/or chondroid differentiation, and displaying *DICER1* variants [58,60], the differential diagnosis with a poorly differentiated SLCT with heterologous rhabdomyoblastic/chondroid element and *DICER1* variant may be difficult. Recently, a comprehensive molecular and epigenetic study of *DICER1*-associated sarcomas has shown that some SLCTS with rhabdomyoblastic differentiation cluster close to *DICER1*-associated sarcomas [58]. Thus, it is possible that some *DICER1*-associated sarcomas could correspond or could develop from a poorly differentiated SLCT, expanding the plasticity of Sertoli cells in sex cord–stromal tumors of the ovary. Several studies are needed to confirm this hypothesis.

## 4. Conclusions

To conclude, since neoplasms within the sex cord–stromal tumor category are sometimes associated, they may represent a continuum (e.g., SCTAT with Sertoli cell tumor or granulosa cell tumor, fibromas/thecomas with minor sex cord elements with, signet ring cell stromal tumor with other sex cord–stromal tumors, gynandroblastoma), for which the final diagnosis often depends on the amount of each component. Therefore, we organized the present review based on the predominant cell morphology (fibromatous/thecomatous cells, steroid cells, follicular cells, Sertoli cells, and sarcomatoid/poorly differentiated cells). However, the morphological criteria are sometimes tricky, especially in the case of luteinized modifications, or sarcomatoid/poorly differentiated tumor. The immunohistochemistry only allows to rule out other ovarian tumor categories. Thus, the identification of diagnostic molecular alterations and a best understanding of the genetics of each entity appears crucial (Table 1). Molecular biology must be used in the case of diagnostic difficulties, to rule out differential diagnoses and to improve patient care. For diagnostic purposes in the sex cord–stromal tumor category, the development of molecular techniques allowing *DICER1* and *FOXL2* variant identification is a minimum requirement, as shown in the diagnostic algorithms proposed herein (Figure 3, Figure 6, Figure 8 and Figure 12). Moreover, pathologists must be aware of the possible associations of sex cord–stromal tumors of the ovary with genetic syndromes (Table 2) [209] in order to suggest genetic investigations in the pathological report. Interestingly, morphological sex cord differentiation has been reported in non-ovarian epithelial (e.g., endometrioid carcinoma of the uterus with sex cord-like formations) [153] and mesenchymal (e.g., uterine tumors resembling ovarian sex cord tumors) neoplasms [210]. However, molecular alternations are different than those usually observed in ovarian sex cord stromal tumors (e.g., no *FOXL2* nor *DICER1* variants). This may suggest that epigenetic and/or microenvironmental factors may be involved in the genesis of sex cord-like formations in these tumors that are not of sex cord stromal origin.

## Figures and Tables

**Figure 1 cancers-15-05864-f001:**
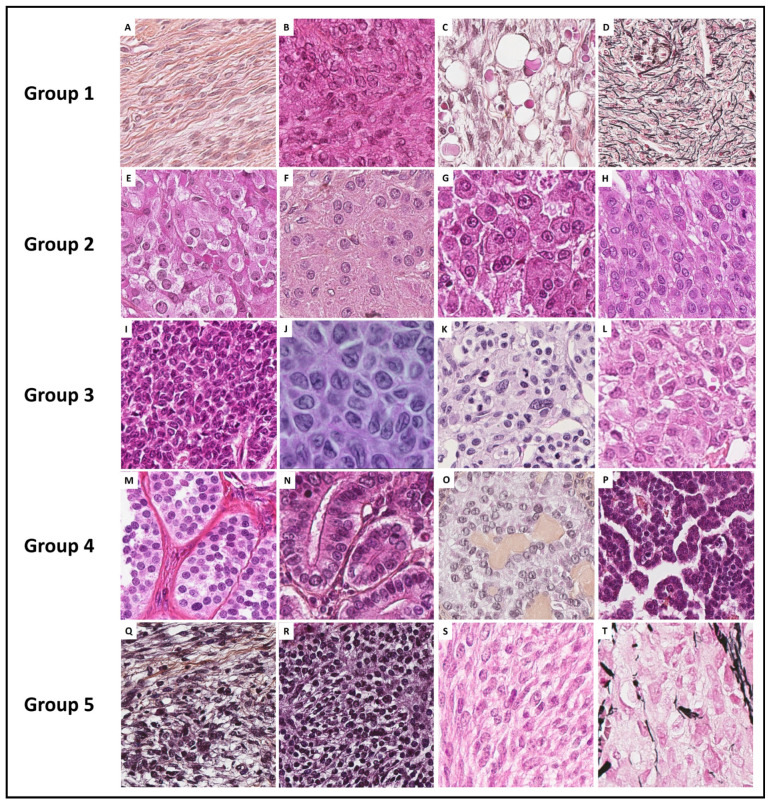
Histopathological illustrations of ovarian sex cord–stromal tumors classified into 5 groups according to predominant cell morphology. Group 1: predominance of fibromatous/thecomatous cells and/or stromal cells of unusual morphology (**A**–**C** [Hematoxylin-eosin-saffron (HES) ×400] and (**D**) [reticuline stain ×400]). Group 2: predominance of Leydig/steroid (**E**–**G** [HES, ×400]) or luteinized cells (**H** [HES, ×400]). Group 3: predominance of follicular cells (**I** [HES, ×400], **J** [HES, ×500], **K** [HES, ×400] and **L** [HES, ×350]). Group 4: predominance of Sertoli cells (**M**–**P** [HES, ×400]). Group 5: predominance of poorly differentiated sex cord cells (**Q**–**S** [HES, ×400] and **T** [reticulin stain, ×400]).

**Figure 2 cancers-15-05864-f002:**
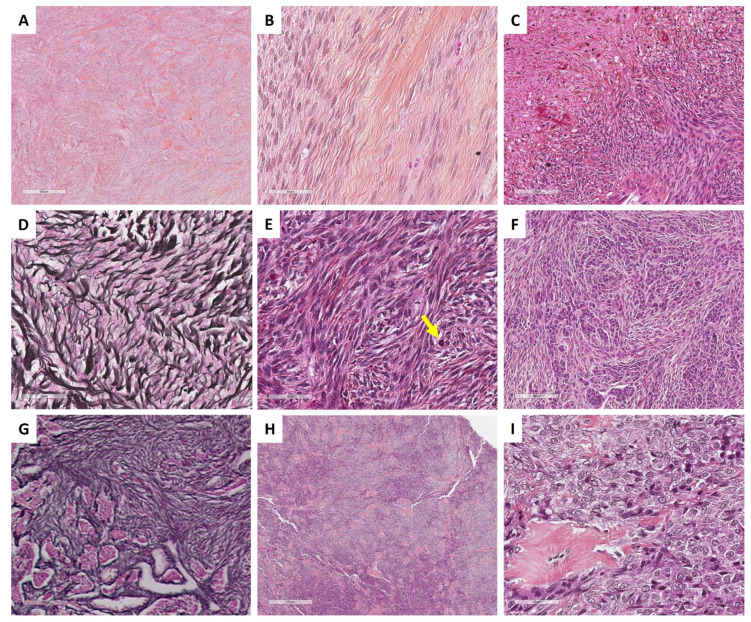
Main microscopic features of ovarian fibromas and thecomas (Group 1). (**A**) (hematoxylin-eosin-saffron [HES], ×40) and (**B**) (HES, ×400), classic fibroma composed of spindle-shaped cells, with scant cytoplasm, elongated or ovoid nuclei without atypia. (**C**) (HES, ×200), necrosis and hemorrhage remodeling. (**D**) (reticulin stain, ×400), dense monocellular reticulin fiber network. (**E**) (HES, ×400), cellular and mitotically active fibroma (yellow arrow: mitosis). (**F**) (HES, ×200) and (**G**) (reticulin stain, ×200), fibroma with minor sex cord element, highlighted by reticulin stain with nests/cords of sex cord elements. (**H**) (HES, ×40) and (**I**) (HES, ×400), thecoma composed of uniform, plump and round, or spindle-shaped cells with pale cytoplasm and indistinct cell membranes.

**Figure 3 cancers-15-05864-f003:**
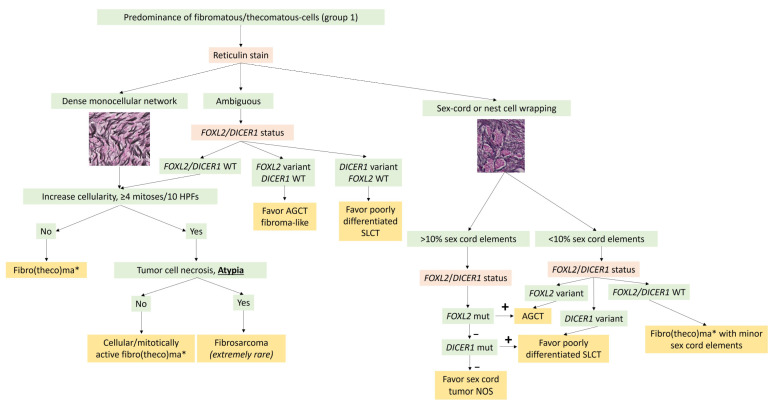
Diagnostic algorithm for Group 1 tumors with predominance of fibromatous/thecomatous cells. * A thecoma component is sometimes associated and usually expresses inhibin and calretinin more intensely than the fibroma component. AGCT: adult granulosa cell tumor; mut: mutation/variant; NOS: not otherwise specified; SLCT: Sertoli–Leydig cell tumor; WT: Wild type.

**Figure 4 cancers-15-05864-f004:**
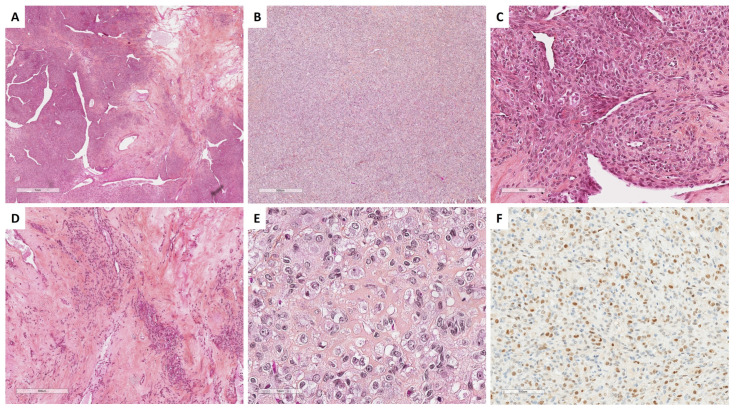
Main microscopic and immunohistochemical features of sclerosing stromal tumor (Group 1). (**A**) (hematoxylin-eosin-saffron [HES], ×20), pseudolobular pattern composed of cellular nodules separated by poorly cellular areas and thin-walled vessels with a hemangiopericytoma-like appearance. (**B**) (HES, ×40), solid/cellular areas may be misleading in a subset of cases. (**C**) (HES, ×200), the tumor is composed of a mix of epithelioid and spindle-shaped cells, alternating with hypocellular areas of fibrosis and sclerosis ((**D**), HES, ×80). (**E**) (HES, ×400), epithelioid cells (luteinized cells) may be predominant, with clear to eosinophilic vacuolated cytoplasm, and may show prominent luteinization. (**F**) (TFE3 immunohistochemistry, ×200), epithelioid cells (luteinized cells) show expression of TFE3.

**Figure 5 cancers-15-05864-f005:**
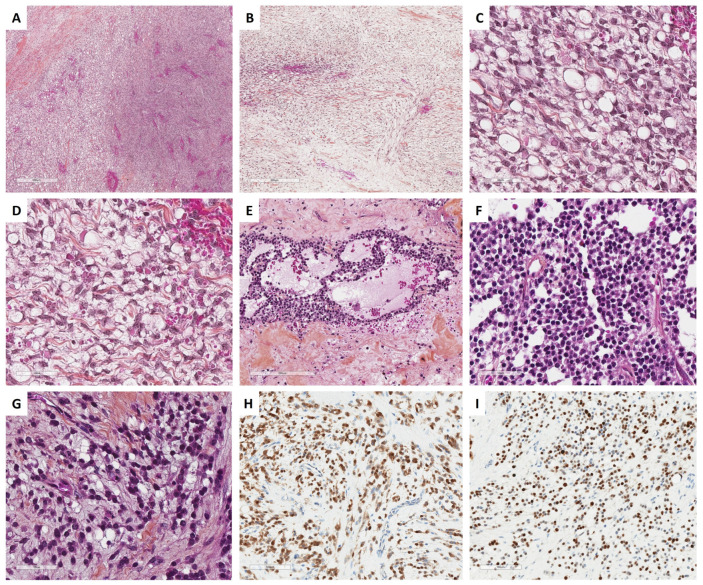
Main microscopic and immunohistochemical features of signet ring cell tumors and microcystic stromal tumors (MCSTs, Group 1). (**A**) (hematoxylin-eosin-saffron [HES], ×40), signet ring cell tumor shows an admixture of fibromatous and signet ring cell areas, sometimes with edematous changes ((**B**), HES, ×80). (**C**) (HES, ×400), signet ring cells have a small, homogenous nuclei, eccentrically located, no to mild atypia, and an empty vacuole, sometimes with hyalin globules, which are degenerating erythrocytes phagocytized by the tumor cells ((**D**), HES, ×400). (**E**) (HES, ×160) and (**F**) (HES, ×400), in contrast, MCSTs exhibit a mix of microcystic, and solid areas composed of epithelioid tumor cells, and fibromatous areas. Cells may show an empty vacuole, resembling signet ring cell tumor ((**G**), HES, ×400); however, in most cases, tumor cells diffusely and intensely express β-catenin ((**H**), ×200) and CyclinD1 ((**I**), ×200).

**Figure 6 cancers-15-05864-f006:**
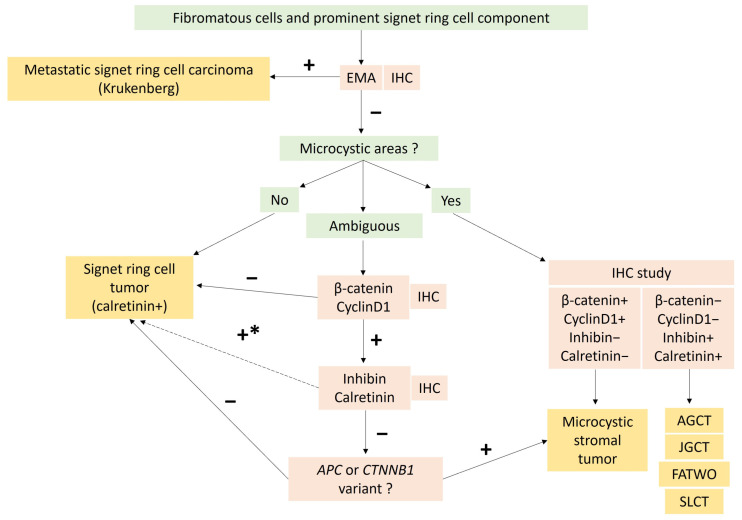
Diagnostic algorithm to differentiate unusual morphological tumor of Group 1: signet ring cell tumor and Microcystic Stromal Tumor (MCST) with signet ring cell changes, according to recently published data [73]. In the case of predominant microcystic and/or follicle-like spaces, combined with intense and diffuse inhibin and calretinin expression, other sex cord tumors and FATWO must be ruled out. * In the case of signet ring cell tumor with diffuse β-catenin and CyclinD1 expression, a search for *ACP* of *CTNNB1* variant may be necessary before concluding signet ring cell stromal tumor [73]. AGCT: Adult granulosa-cell tumor; EMA: Epithelial membrane antigen; FATWO: Female adnexial tumor of Wolfian origin; IHC: Immunohistochemistry; JGCT: Juvenile granulosa-cell tumor; SLCT: Sertoli Leydig-cell tumor.

**Figure 7 cancers-15-05864-f007:**
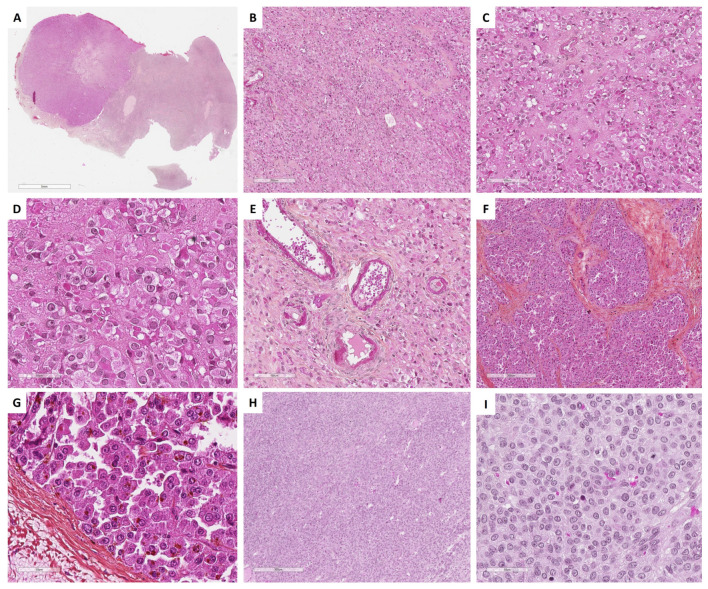
Main microscopic features of Group 2 tumors: Steroid and luteinized cells. (**A**) (hematoxylin-eosin-saffron [HES], ×5), hilar location of a Leydig cell tumor. (**B**) (HES, ×100), Leydig cell tumor: diffuse cell growth. (**C**) (HES, ×400), clustering of nuclei, separated by eosinophilic nuclear-free areas. (**D**) (HES, ×400), Leydig cells with abundant eosinophilic cytoplasms and round nuclei with a central nucleolus. (**E**) (HES, ×200), eosinophilic fibrinoid material within vessel walls. (**F**) (HES, ×80), steroid cell tumor not otherwise specified (NOS) with diffuse and lobulated architecture. (**G**) (HES, ×400), tumor cells of a steroid cell tumor NOS, displaying abundant eosinophilic and granular cytoplasm, lipochrome pigment, round nuclei with a central nucleolus. (**H**) (HES, ×200), luteinized adult granulosa cell tumor, with diffuse architecture. (**I**) (HES, ×200), luteinized cells of a luteinized adult granulosa cell tumor, displaying abundant grey cytoplasm, as seen in some thecomas, with round nuclei that have lost typical granulosa cell nucleolus features.

**Figure 8 cancers-15-05864-f008:**
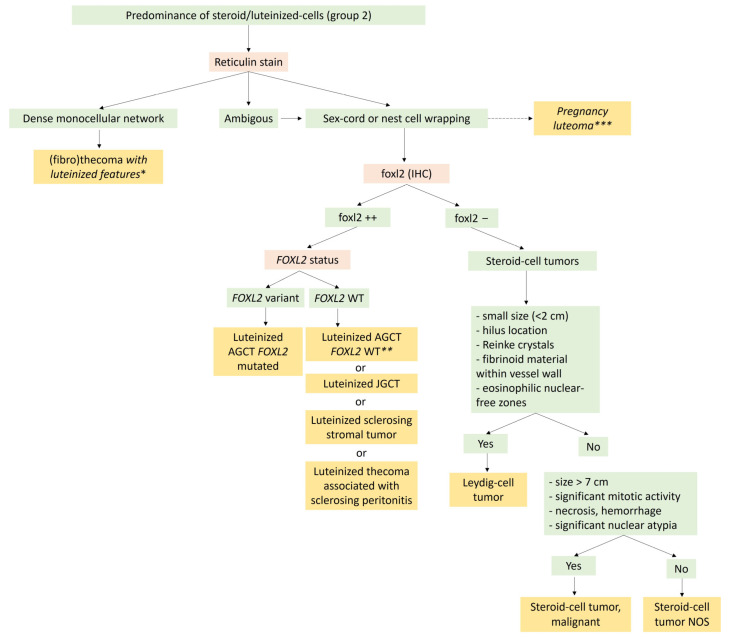
Diagnostic algorithm for Group 2 tumors with a predominance of steroid/luteinized cells. * Rule out the luteinized thecoma associated with sclerosing peritonitis, which is almost always bilateral, with ascites, and bowel obstruction because of the sclerosing peritonitis, and for which the reticulin stain also surrounds the cluster of luteinized cells. ** In the absence of the *FOXL2* variant, this diagnosis must be performed only if true sex cord or nest cells are visible on the reticulin stain, and after ruling out other neoplasms with luteinized cells, such as sclerosing stromal tumors and luteinized thecoma associated with sclerosing peritonitis. *** In the case of pregnancy or immediate post-partum. AGCT: Adult granulosa-cell tumor; JGCT: Juvenile granulosa-cell tumor; NOS: Not other specified.

**Figure 9 cancers-15-05864-f009:**
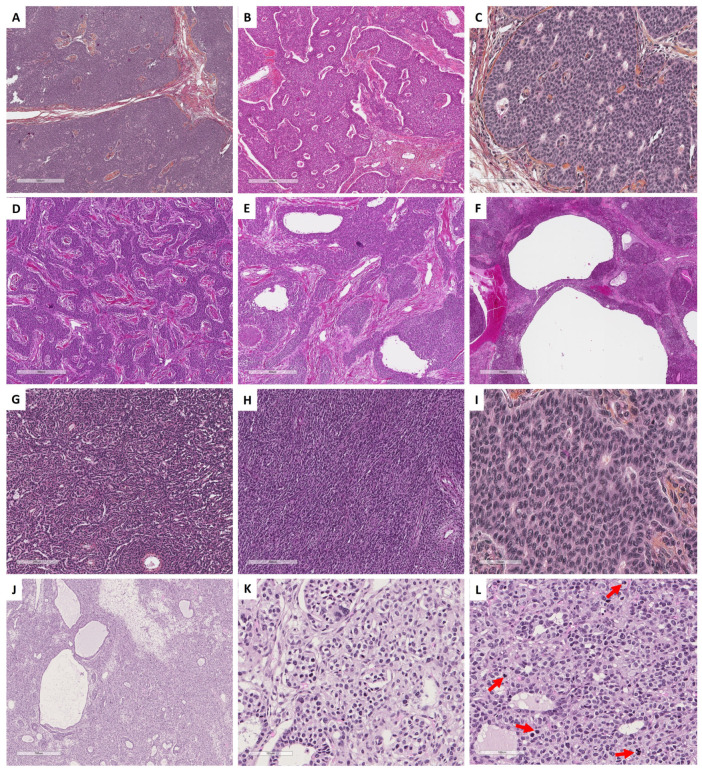
Main microscopic features of Group 3 tumors: predominance of follicular cells. (**A**) (hematoxylin-eosin-saffron [HES], ×45) and (**B**) (HES, ×60), diffuse and insular architectures of adult granulosa cell tumor. (**C**) (HES, ×200), microfollicular with Call–Exner bodies of adult granulosa cell tumor. (**D**) (HES, ×40), cord/trabecular pattern of adult granulosa cell tumor. (**E**) (HES, ×80) and (**F**) (HES, ×30), macrofollicular pattern of adult granulosa cell tumor. (**G**) (HES, ×100), moire-silk pattern of adult granulosa cell tumor. (**H**) (HES, ×80), sarcomatoid pattern of adult granulosa cell tumor. (**I**) (HES, ×400), typical nuclear grooves of adult granulosa cell tumor. (**J**) (HES, ×30), juvenile granulosa cell tumor displays a diffuse, lobulated, and follicle-like architecture. (**K**) (HES, ×200) and (**L**) (HES, ×200), tumor cells have abundant eosinophilic cytoplasm, and round vesicular to hyperchromatic nucleus that typically lack the nuclear grooves observed in the adult granulosa cell tumor, with numerous mitoses (red arrows).

**Figure 10 cancers-15-05864-f010:**
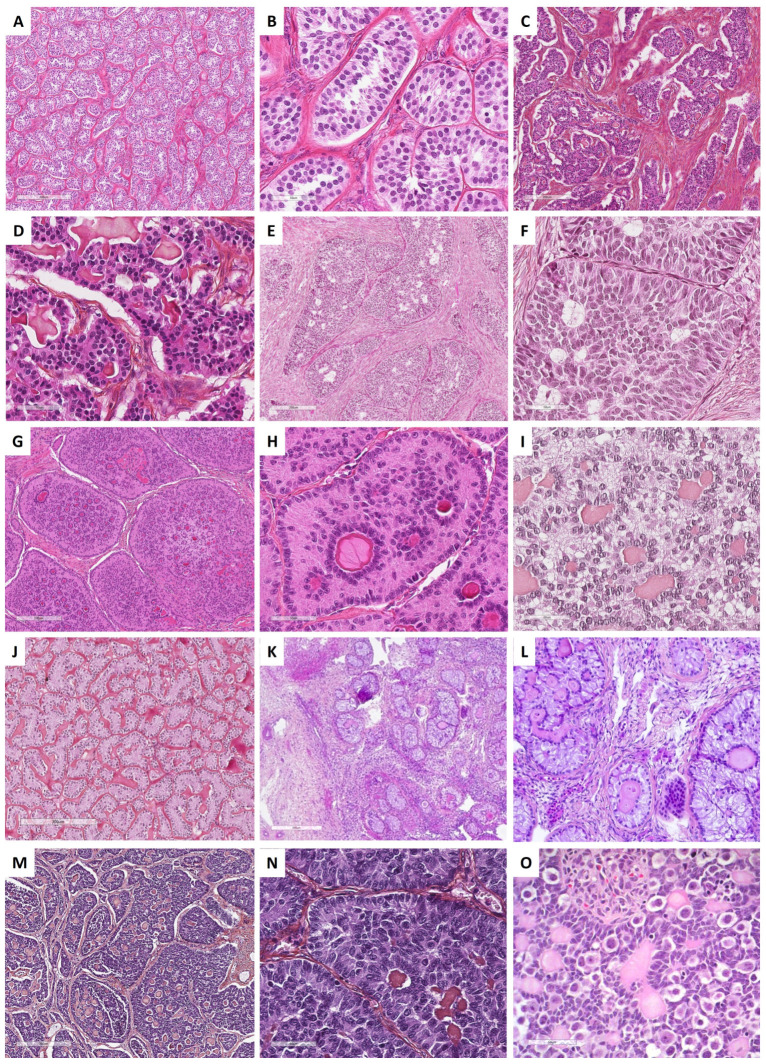
Main microscopic features of Pure Sertoli cell tumors and Sex cord tumors with annular tubules (SCTAT) and differential diagnoses (Group 4). (**A**) (hematoxylin-eosin-saffron [HES], ×80), Sertoli cell tumors are typically composed of hollow or solid tubules resembling prepubertal testicular tubules. (**B**) (HES, ×400), Sertoli cells have moderate amount of clear vacuolated lipid-rich to brightly eosinophilic cytoplasm, with small round nuclei with a central nucleolus without atypia, and a low mitotic count. (**C**) (HES, ×100) and (**D**) (HES, ×400), ovarian carcinoids may display insular, tubular, trabecular, and acinar architectures, with monomorphic cells composed of a centrally-located, round to oval nuclei, salt-and-pepper chromatin, and pink cytoplasm, resembling sex cord–stromal tumors. (**E**) (HES, ×100) and (**F**) (HES, ×400), some endometrioid carcinomas may show sex cord–stromal features characterized by nested and corded arrangements, and cells resembling sex cord tumor cells (e.g., Sertoli cells, adult granulosa cells). (**G**) (HES, ×100), SCTAT are composed of well-circumscribed round or complex nests of Sertoli cells, tubule-forming, that encircle hyaline basement membrane-like material with a round or oval shape. (**H**) (HES, ×400) and (**I**) (HES, ×400), tumor cells have abundant clear, lipid-rich, vacuolated to pale eosinophilic cytoplasms, with small round to oval nuclei, with single small nucleoli, but without atypia, and with a low mitotic count. (**J**) (HES, ×100), in SCTAT, areas of Sertoli tubules may be observed. (**K**) (HES, ×40) and (**L**) (HES, ×200), in SCTAT associated with Peutz–Jeghers syndrome, simple nests of Sertoli cells with are usually distributed within the ovarian fibromatous stroma, surrounding hyaline basement membrane-like material. (**M**) (HES, ×80) and (**N**) (HES, ×400), adult granulosa cell tumor with SCTAT-like features (abundant hyalinized basement membrane-like material within Call–Exner bodies). (**O**) (HES, ×400), gonadoblastoma are composed of sex cord component surrounding hyaline basement membrane-like material, mixed with a germ cell component.

**Figure 11 cancers-15-05864-f011:**
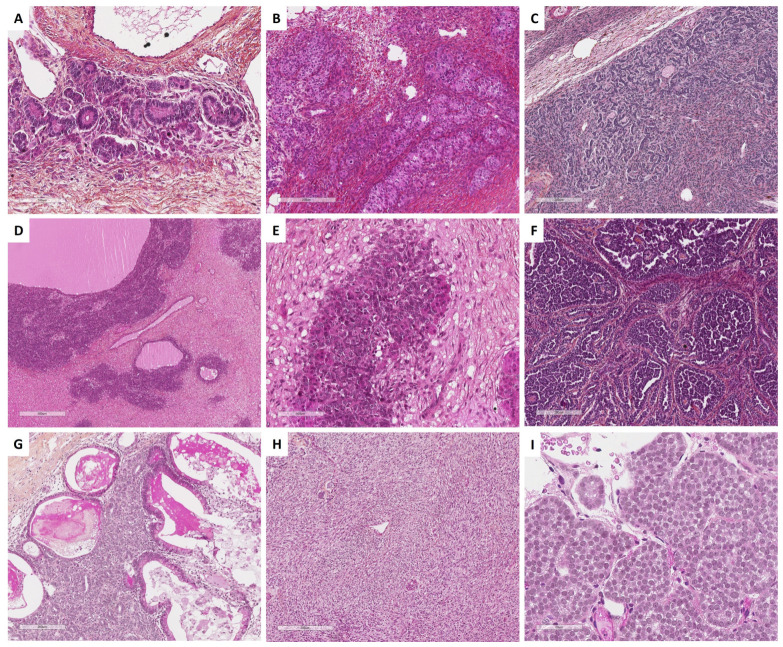
Main microscopic features of Sertoli–Leydig cell tumors (SLCT; Group 4). (**A**) (hematoxylin-eosin-saffron [HES], ×200) and (**B**) (HES, ×120), well-differentiated SLCT. (**C**) (HES, ×100), moderately differentiated SLCT. (**D**) (HES, ×40) and (**E**) (HES, x200), poorly differentiated SLCT. (**F**) (HES, ×100), retiform component in a poorly differentiated SLCT. (**G**) (HES, ×100), heterologous element: benign intestinal type epithelium. (**H**) (HES, ×80), heterologous element: rhabdomyosarcoma. (**I**) (HES, ×400), heterologous element: carcinoid tumor.

**Figure 12 cancers-15-05864-f012:**
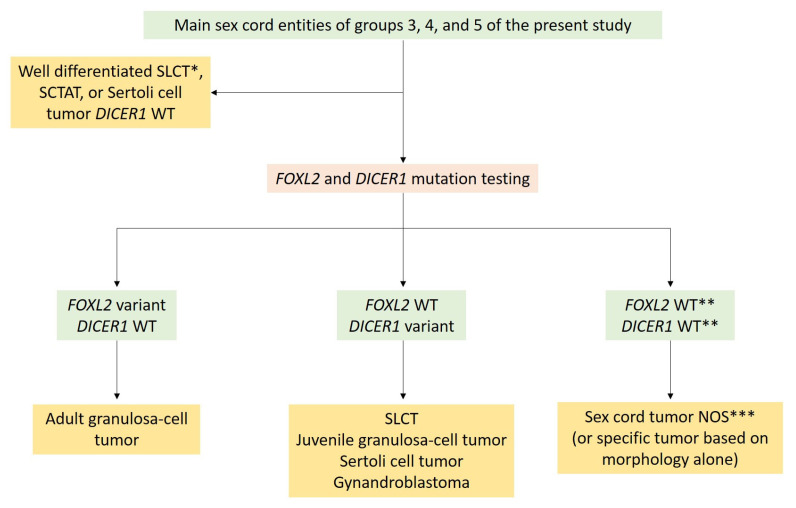
Simplified diagnostic algorithm between sex cord tumors based on the main molecular alterations. * Recent data tend to separate well-differentiated SLCT as a distinct entity from other SLCT [6]. ** In the case of poorly differentiated sex cord tumor with FOXL2 and DICER1 WT status. *** Although other moderately/poorly differentiated neoplasms could belong to this category of FOXL2/DICER1 WT sex cord tumors (such as adult granulosa cell tumor FOXL2 WT, moderately/poorly differentiated SCLT DICER1 WT), some features are usually present (well-differentiated areas and “coffee bean” nuclei for adult granulosa tumor, Leydig cells and Sertoli tubes for SLCT), allowing the classification as adult granulosa cell tumor or moderately/poorly differentiated SLCT, respectively. Otherwise, the neoplasm must be classified as sex cord–stromal tumor NOS. NOS: Not otherwise specified; SCTAT: Sex cord tumor with annular tubules; SLCT: Sertoli–Leydig cell tumor; WT: Wild type.

**Table 1 cancers-15-05864-t001:** Main molecular abnormalities associated with stromal and sex cord entities.

Stromal and Sex Cord Entities	Molecular Abnormalities
Fibroma/Thecoma	- Trisomy or tetrasomy of chromosome 12 (63%). - Gain of chromosomes 9, 9q (50%), 18 (20%), and 21 (20%). - Loss of heterozygosity (LOH) at 9q22.3 (proximal *PTCH1*) in 25%. - LOH at 9q22.3 (67%) and at 19p13.3 (*STK11*, 50%) in cellular fibromas. - *PTCH1* variant, associated with *SMARCA4* variant in one case. - *IDH1* variant associated with Ollier disease in one case. - No *FOXL2* or *DICER1* variant *.
Fibrosarcoma	- Tetrasomy 12. - Trisomy 12 and 8. - Amplification of large regions in chromosomes 1, 2, 7, 8, 17. - Amplification of *MYC.* - Deletion of *TP53*. - *DICER1* and *NF1* variants. - No *FOXL2* or *DICER1* variant **.
Sclerosing stromal tumor	- Trisomy 7 and 12. ***GLI2* rearrangements (81%):** - *FHL2::GLI2* fusion (65%) - *DYNLL1::GLI2* fusion, or other *GLI2* fusion with a gene partner not yet identified (15%)
Signet ring cell tumor ***	- No variant of *FOXL2*
Microcystic stromal tumor	- **Mutually exclusive *CTNNB1* (70.4%) and *APC* variants.** - Other gene variants identified: *RET, FANCA2, KRAS.* - No *FOXL2* or *DICER1* variant.
Leydig cell tumor/Steroid cell tumor NOS/malignant	NA
Luteinized thecoma associated with sclerosing peritonitis	- *MGAT5B::NCOA3* fusion in one case.
Adult granulosa cell tumor	- ***FOXL2* variant (>95%).** - *TERT* promoter alteration and *TP53* variants - Gains of chromosomes 12 and 14 (30%), and loss of chromosome 22 (40–50%).
Juvenile granulosa cell tumor	- ***DICER1* variant** (6–23%) - Variants of: *AKT1* (29–60%), *KMT2C* (52%), *ARID1A* (39%), *KMT2D* (35%), *BRCA2* (35%), *GNAS* (30%), *ATM* (26%), *BRCA1* (23%), *POLE* (23%), *MLH1* (19%), *TERT* (19%), *MSH6* (10%), *IDH1/2.* - *TERT* rearrangements (13%). - No *FOXL2* variant ****.
Sertoli–Leydig cell tumor (moderately to poorly differentiated only)	- ***DICER1* variant (63%)** - CDC27 variant (52.6%) - MUC22 variant (21.1%)
Pure Sertoli cell tumor	- ***DICER1* variant (63%)** - ***STK11* germline variant**
Sex cord tumor with annular tubules	- **Germline *STK11* variant** or **19p13.3 LOH**
Gynandroblastoma	- ***DICER1* variant (19–40%)** - No variant of *FOXL2* or *AKT1*

* Although thecomas harboring *FOXL2* somatic variant have been reported, the peritoneal recurrence as an adult granulosa cell tumor suggests that the initial “thecoma” might have in fact represented a luteinized adult granulosa cell tumor. ** Although fibrosarcoma has been reported with *DICER1* variant [53], considering the recent molecular findings in the literature, this neoplasm would preferentially be classified as *DICER1*-associated sarcoma [58,59,60]. *** Although *CTNNB1* variant has been reported in this neoplasm, it is not unanimously accepted, and some authors recommend to rather classify those tumors with *CTTNB1* variant as a morphological variant of microcystic stromal tumor. **** Juvenile granulosa cell tumors with *FOXL2* variant have rarely been reported. However, it is unclear if these cases were true juvenile granulosa cell tumors or adult granulosa-cell tumors. LOH: Loss of heterozygosity. NA: data not available; NOS: Not otherwise specified. In bold type: main diagnostic molecular alterations.

**Table 2 cancers-15-05864-t002:** Syndromes associated with stromal and sex cord tumors.

Syndromes	Sex Cord–Stromal Tumor Entities
Gorlin syndrome	Fibroma, fibrosarcoma
Ollier disease	Fibroma, juvenile granulosa cell tumor
Maffucci syndrome	Fibrosarcoma, juvenile granulosa cell tumor
Familial adenomatous polyposis, 5q deletion syndrome (including *APC* deletion)	Microcystic stromal tumor
Peutz–Jeghers syndrome	Sex cord tumor with annular tubules, pure Sertoli cell tumor
DICER1 syndrome	Sertoli–Leydig cell tumor, (*fibrosarcoma*) *, juvenile granulosa cell tumor, gynandroblastoma
Von Hippel–Lindau Syndrome	Steroid cell tumor NOS
Tuberous Sclerosis	Juvenile granulosa cell tumor
Beckwith–Wiedmann syndrome	Juvenile granulosa cell tumor

* Although the initial diagnosis of fibrosarcoma with *DICER1* variant was made [53], considering the recent molecular findings in the literature, this neoplasm would preferentially be classified as *DICER1*-associated sarcoma [58,59,60].

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
