# Peer review of "Relevance of Molecular Pathology for the Diagnosis of Sex Cord–Stromal Tumors of the Ovary: A Narrative Review"

_cancers, 2023, doi:10.3390/cancers15245864_

Round 1

Reviewer 1 Report

Comments and Suggestions for Authors

This is a narrative review on the Relevance of molecular pathology for the diagnosis of sex cord-stromal tumors of the ovary. The authors give a complete and exhaustive overview of the diagnosis of the different types of sex cord-stromal tumors of the ovary,Underline the importance of molecular pathology. 

I suggest to implement the references also with this recent revision on UTROSC https://doi.org/10.3390/jcm12227131 and to add in the discussion also the possibilities similarities with uterine tumour resembling ovarian sex cordon 

Author Response

Response to reviewers’ comments (cancers-2748442)

We would like to thank the editor and the reviewers for their constructive comments that have helped us improve the quality of the manuscript and for the time they dedicated towards reading our work. You will find below a point-by-point response to the comments and a revised version of the manuscript (cancers-2748442), entitled “Relevance of molecular pathology for the diagnosis of sex cord-stromal tumors of the ovary: a narrative review” in which all changes have been highlighted in yellow.

Reviewer 1:

Comment #1: This is a narrative review on the Relevance of molecular pathology for the diagnosis of sex cord-stromal tumors of the ovary. The authors give a complete and exhaustive overview of the diagnosis of the different types of sex cord-stromal tumors of the ovary, Underline the importance of molecular pathology.

I suggest to implement the references also with this recent revision on UTROSC https://doi.org/10.3390/jcm12227131 and to add in the discussion also the possibilities similarities with uterine tumour resembling ovarian sex cordon

Answer #1:

We thank this reviewer for his comment. As requested, the suggested reference has been added at the end of the conclusion, to discuss the possibility of "sex cord-like" morphology in non-ovarian neoplasms.

“Interestingly, morphological sex-cord differentiation has been reported in non-ovarian epithelial (e.g., endometrioid carcinoma of the uterus with sex cord-like formations) [159] and mesenchymal (e.g., uterine tumors resembling ovarian sex-cord tumors) neoplasms [220]. However, molecular alternations are different than those usually observed in ovarian sex-cord stromal tumors (e.g., no FOXL2 nor DICER1 variants). This may suggest that epigenetic and/or microenvironnemental factors may be involved in the genesis of sex cord-like formations in these tumors that are not of sex cord stromal origin”.

Reviewer 2 Report

Comments and Suggestions for Authors

In this manuscript, Alexis Trecourt and colleagues review in detail the diagnosis of sex-cord stromal tumors of the ovary. They integrate histological information with molecular pathology information generating an amazing piece of work of high value for pathologists and researchers of the field. I strongly support its publication in Cancers.

I do have a minor suggestion. While discussing adult granulosa cell tumors, the authors mention that "FOXL2 is known as a marker of ovarian differentiation and has a role in the proliferation and differentiation of granulosa cells." Additionally, they state, "It is not clear whether the presence of a FOXL2 variant has a poor prognostic impact or not" (page 21 of 45, lines 783-4 and 787-8). Recently, Llano et al. (DOI: 10.1158/0008-5472.CAN-22-1880) demonstrated that FOXL2-C134W drives the development of granulosa cell tumors in a mouse model, suggesting its oncogenic role in this type of cancer. I recommend that the authors consider incorporating this information and rephrase the relevant portion of this paragraph.

Additionally, there are a couple of typos:

Line 1182: "nor" instead of "and."

Line 1297: "Syndromes" instead of "syndromes."

Author Response

Response to reviewers’ comments (cancers-2748442)

We would like to thank the editor and the reviewers for their constructive comments that have helped us improve the quality of the manuscript and for the time they dedicated towards reading our work. You will find below a point-by-point response to the comments and a revised version of the manuscript (cancers-2748442), entitled “Relevance of molecular pathology for the diagnosis of sex cord-stromal tumors of the ovary: a narrative review” in which all changes have been highlighted in yellow.

Reviewer 2:

Comment #1:

In this manuscript, Alexis Trecourt and colleagues review in detail the diagnosis of sex-cord stromal tumors of the ovary. They integrate histological information with molecular pathology information generating an amazing piece of work of high value for pathologists and researchers of the field. I strongly support its publication in Cancers.

I do have a minor suggestion. While discussing adult granulosa cell tumors, the authors mention that "FOXL2 is known as a marker of ovarian differentiation and has a role in the proliferation and differentiation of granulosa cells." Additionally, they state, "It is not clear whether the presence of a FOXL2 variant has a poor prognostic impact or not" (page 21 of 45, lines 783-4 and 787-8). Recently, Llano et al. (DOI: 10.1158/0008-5472.CAN-22-1880) demonstrated that FOXL2-C134W drives the development of granulosa cell tumors in a mouse model, suggesting its oncogenic role in this type of cancer. I recommend that the authors consider incorporating this information and rephrase the relevant portion of this paragraph.

Answer #1:

We thank this reviewer for his comment. Indeed, these sentences were ambiguous and have been reworded taking into account the reference suggested by this reviewer.

FOXL2 is known as a marker of ovarian differentiation and has a role in the proliferation and differentiation of granulosa cells [132, 133], and its alteration reduce fertility in mouse model [134]. Recently, in vivo studies demonstrated that the somatic variant c.402C>G (p.C134W) of FOXL2 was necessary and sufficient to trigger the genesis of adult granulosa cell tumor [134]. This alteration led to dysregulation of the TGFβ pathway signaling, which is consistent with previously reported data [133].”

Comment #2:

Additionally, there are a couple of typos:

Line 1182: "nor" instead of "and."

Line 1297: "Syndromes" instead of "syndromes."

Answer #2:

This has been revised accordingly.